

# Modeling organic aerosol concentrations and properties during winter 2014 in the northwestern Mediterranean region

Mounir Chrit[1], Karine Sartelet[1], Jean Sciare[2,3], Marwa Majdi[1,5], José Nicolas[2], Jean-Eudes Petit[2,4], and François Dulac[2]

[1]CEREA, joint laboratory Ecole des Ponts ParisTech - EDF R&D, Université Paris-Est, 77455 Champs sur Marne, France.

[2] LSCE, CNRS-CEA-UVSQ, IPSL, Université Paris-Saclay, Gif-sur-Yvette, France

[3] EEWRC, The Cyprus Institute, Nicosia, Cyprus

[4] INERIS, Parc Technologique ALATA, 60550 Verneuil-en-Halatte, France

[5] Laboratoire de Métérologie Dynamique (LMD)-IPSL, Sorbonne Université, CNRS UMR 8539, Ecole Polytechnique, Paris, France

*Correspondence to:* Mounir CHRIT (mounir.chrit@enpc.fr)

**Abstract.** Organic aerosols are measured at a remote site (Ersa) on Corsica Cape in the northwestern Mediterranean basin during the Chemistry-Aerosol Mediterranean Experiment (CharMEx) winter campaign of 2014, when high organic concentrations from anthropogenic origin are observed. This work aims at representing the observed organic aerosol concentrations and properties (oxidation state) using the air-quality model Polyphemus with a surrogate approach for secondary organic aerosol

(SOA) formation. Because intermediate/semi-volatile organic compounds (I/S-VOC) are the main precursors of SOA at Ersa during the winter 2014, different parameterizations to represent the emission and ageing of I/S-VOC were implemented in the chemistry-transport model of the air-quality platform Polyphemus (different volatility distribution emissions, single-step oxidation vs multi-step oxidation within a Volatility Basis Set framework, inclusion of non-traditional volatile organic compounds NTVOC). Simulations using the different parameterizations are compared to each other and to the measurements (concentra-

tion and oxidation state). The high observed organic concentrations are well reproduced whatever the parameterizations. They are slightly under-estimated with most parameterizations, but they are slightly over-estimated when the ageing of NTVOC is taken into account. The volatility distribution at emissions influences more strongly the concentrations than the choice of the parameterization that may be used for ageing (single-step oxidation vs multi-step oxidation), stressing the importance of an accurate characterization of emissions. Assuming the volatility distribution of sectors other than residential heating to be the

same as residential heating may lead to a strong under-estimation of organic concentrations. The observed organic oxidation and oxygenation states are strongly under-estimated in all simulations, even when a recently developed parameterization for modeling the ageing of I/S-VOC from residential heating is used. This suggests that uncertainties in the ageing of I/S-VOC emissions remain to be elucidated, with a potential role of organic nitrate from anthropogenic precursors and highly oxygenated organic molecules.




# 1 Introduction

Organic aerosols (OA) are one of the main compound of submicron particulate matter (PM$_1$) (Jimenez et al., 2009). Their primary fraction originates from combustion sources, such as traffic and residential heating. However, large uncertainties remain regarding their emissions (Gentner et al., 2017; Shrivastava et al., 2017). POA has been considered as non-volatile in emissions inventories and chemistry-transport models (CTMs); however, recent studies have provided clear evidences that a large portion of POA emissions partition between the gas and the particle phases (Robinson et al., 2007). Organic species that compose OA are often classified depending on their volatility: intermediate volatility organic compounds (IVOC) (with saturation concentration C$^*$ in the range 10$^4$-10$^6$ $\mu$g m$^{-3}$), semi-volatile organic compounds (SVOC) (with saturation concentration C$^*$ in the range 0.1-10$^4$ $\mu$g m$^{-3}$), or low-volatility organic compounds (LVOC) (with saturation concentration C$^*$ lower than 0.1 $\mu$g m$^{-3}$) (Lipsky and Robinson, 2006; Grieshop et al., 2009; Huffman et al., 2009; Cappa and Jimenez, 2010; Fountoukis et al., 2014; Tsimpidi et al., 2010; Woody et al., 2016; Ciarelli et al., 2017b, a).

OA originates not only from the partitioning of POA between the gas and the particle phases, but also from secondary aerosol formation (SOA) through the gas-to-particle partitioning of the oxidation products of biogenic and anthropogenic volatile organic compounds (VOC) and intermediate and semi volatile organic compounds (I/S-VOC). The main biogenic VOC precursors are terpenes ($\alpha$-pinene, $\beta$-pinene, limonene, humulene) and isoprene (Shrivastava et al., 2017), while the main anthropogenic VOC are aromatics (e.g. toluene, xylenes) (Dawson et al., 2016; Gentner et al., 2017).

Available measurements and modeling studies are useful to elucidate the composition and origin of OA in different seasons (Couvidat et al., 2012; Hayes et al., 2015; Canonaco et al., 2015; Chrit et al., 2017; Ciarelli et al., 2017b). Indeed, over the Mediterranean region, the oxidation of biogenic VOC may dominate the formation of OA during the summer (El Haddad et al., 2013; Minguillón et al., 2016; Chrit et al., 2017). Chrit et al. (2017) found that I/S-VOC emissions do not influence much the concentrations of OA in summer over the Mediterranean region, but biogenic SOA prevail. Because biogenic emissions are low in winter, Canonaco et al. (2015) demonstrated a clear shift in the SOA origin between summer and winter during a measurement campaign from February 2012 to February 2013 conducted in Zürich using the Aerosol Chemical Speciation Monitor (ACSM, Ng et al. (2011)) measurements. This last study notably highlights the importance of biogenic VOC emissions and biogenic SOA production in summer, and the importance of residential heating in winter. Ciarelli et al. (2017a) performed a source apportionment study at the European scale and revealed that residential combustion (mainly related to wood burning) contributed around 60-70% to SOA formation during the winter whereas non-residential combustion and road-transportation sector contributed about 30-40% to SOA formation. Moreover, residential heating can also be a source of POA, which may make up a large fraction (20% to 90%) of the submicron particulate matter in winter (Murphy et al., 2006; May et al., 2013d; Shrivastava et al., 2017).

Modeling OA concentrations in winter is challenging, because it involves mostly the characterization of I/S-VOC emissions and ageing. Standard gridded emission inventories, such as those of the European Monitoring and Evaluation Programme (EMEP, www.emep.int) over Europe, do not yet include I/S-VOC emissions, and their emissions are still highly uncertain. For example, Denier van der Gon et al. (2015) estimated that emissions from residential wood combustion were under-estimated



by a factor 2-3 in the 2005 EUCAARI inventory. As an indirect method to account for the missing organic emissions in the absence of precise emission inventories, numerous modeling studies estimate the I/S-VOC emissions from POA emissions (Couvidat et al., 2012; Bergström et al., 2012; Koo et al., 2014; Zhu et al., 2016; Ciarelli et al., 2017a) or more recently from VOC emissions (Zhao et al., 2015, 2016; Ots et al., 2016; Murphy et al., 2017). A ratio of I/S-VOC/POA of 1.5 has been used

in several air quality studies (Bergström et al., 2012; Koo et al., 2014; Zhu et al., 2016; Ciarelli et al., 2017a). For example, Zhu et al. (2016) simulated the particle composition over Greater Paris during the winter MEGAPOLI campaign and they found that simulated OA agreed well with observed OA when gas-phase I/S-VOC emissions are estimated using a ratio I/S-VOC/POA of 1.5, as derived following the measurements at the tailpipe of vehicles representative of the french fleet (Kim et al., 2016). However, various ratios are used to better fit the measurements. For example, over Europe, Couvidat et al. (2012) used a ratio

I/S-VOC/POA of 4 but also of 6 in a sensitivity simulation to better fit the observed OA concentrations in winter. Koo et al. (2014) used a ratio IVOC/POA of 1.5 but also of 3 in their high IVOC emission scenario.

    The atmospheric evolution (also known as ageing) of I/S-VOC as well as their impacts on atmospheric OA concentrations remain poorly characterized (Murphy et al., 2006) and deserve a better understanding. A widely used approach to model the ageing of I/S-VOC in CTMs is the volatility basis set (VBS) approach (Donahue et al., 2006). I/S-VOC are divided into sev-

eral classes of volatility where each class is represented by a surrogate. When oxidized by the hydroxyl radical, it leads to the formation of surrogates of lower volatility classes. This approach tends to lead to an overestimation of simulated organic concentrations (Cholakian et al., 2017) if fragmentation is not considered (formation of high volatility surrogates during the oxidation). Although the one-dimensional basis set 1-D VBS accounts for the volatility of the surrogates, it does not allow the representation of varying oxidation levels of OA. The more powerful prognostic tool to date, bi-dimensional VBS approach

(2D-VBS), although it is computationally burdensome, describes the ageing of I/S-VOC using not only the volatility property ($C^*$) but also the oxidation level (the oxygen-to-carbon ratio O:C), taking into account two competing processes: functionalization and fragmentation (Donahue et al., 2012). Koo et al. (2014) developed a 1.5-D ageing VBS-type scheme that accounts for fragmentation, functionalization and multigenerational ageing, and that represents both the volatility and the oxidation properties of the surrogates. When oxidized by a hydroxyl radical, each surrogate leads to the formation of more oxidized and

less volatile surrogates with a reduced carbon number. Functionalization and fragmentation are implicitly taken into account in this approach, because of the increase of the oxygen number and the decrease of the carbon number of the surrogates formed. The 1.5-D VBS module is implemented within two widely used CTMs namely CAMx(ENVIRON, 2011) and CMAQ (Byun and Ching, 1999). Couvidat et al. (2012, 2013b, 2017) and Zhu et al. (2016) used a simplified ageing scheme with 3 volatility bins. When oxidized by the hydroxyl radical, each surrogate forms a less volatile and more oxidized surrogate, that does not

undergo multigenerational ageing. This simplified ageing scheme is implemented in the two widely used CTMs Polyphemus (Chrit et al., 2017) and Chimere (Couvidat et al., 2017). In winter, when anthropogenic emissions impact the most air quality, anthropogenic emissions such as toluene and xylenes may also form SOA, although they may be less efficient than I/S-VOC (Couvidat et al., 2013a). To take into account emissions and ageing of anthropogenic VOC that are usually not considered in CTMs (phenol, naphtalene, m-,o-,p- cresol, etc.), Ciarelli et al. (2017b) modified the approach of Koo et al. (2014) by adding

non traditional VOC (NTVOC) that have a limit saturation concentration between VOC and IVOC.





The oxidation level of OA is important, because it is indicative of the degree of hygroscopicity, surface tension (Jimenez et al., 2009), and radiative property of the OA in addition to its ability to act as cloud condensation nuclei (CCN) over the Mediterranean (Jimenez et al., 2009; Duplissy et al., 2011; Wong et al., 2011). Chrit et al. (2017) showed that, in summer in the western Mediterranean region, OA is highly oxidized and oxygenated. The CTM Polyphemus/Polair3d used in their study does represent this high oxidation level of OA after adding to the model formation processes of highly oxidized species (autoxidation) and organic nitrate formation.

Although the organic matter to organic carbon ratio (OM:OC) was first believed to lie between 1.2 and 1.4 (Grosjean and Friedlander, 1975), recent studies (Turpin and Lim, 2001; El-Zanan et al., 2005) show that OM:OC is rather close to 1.6 for urban aerosols and 2.1 for non urban aerosols. Zhang et al. (2005a) developed an algorithm to deconvolve the mass spectra of OA obtained with an Aerodyne$^{TM}$ Aerosol Mass Spectrometer (AMS) in order to estimate the mass concentrations of hydrocarbon-like and oxygenated organic aerosols (HOA and OOA). The mass of HOA represents primary sources, with a OM:OC ratio close to 1.2 and O:C ratio close to 0.1, while the mass of OOA represents secondary sources (aged and oxygenated) with a OM:OC ratio close to 2.2 and O:C ratio close to 1 (Aiken et al., 2008). Using this technique, Zhang et al. (2005b) found an average OM:OC ratio of 1.8 in Pittsburgh in September. Over Europe, Crippa et al. (2014) found that secondary OA is dominant in the OA fraction, with primary sources contributing to less than 30% to the total mass fraction. Xing et al. (2013) measured a ratio OM:OC ratio over 14 cities throughout China and found that in summer, OM:OC is nearly $1.75 \pm 0.13$, while the ratio is lower in winter ($1.59 \pm 0.18$). The OM:OC ratio is lower during winter due to the slow oxidation process owing to the low temperatures in addition the low biogenic contribution to OA mass during winter. At Ersa, over the Mediterranean during the summer, Chrit et al. (2017) found high OM:OC and O:C ratios (2.5 and 1 respectively). They are due to aged biogenic OA, which Chrit et al. (2017) were able to represent by adding the formation of extremely low-volatility species and organic nitrate to the model and by considering the formation of organosulfate.

Quantifying the effect of I/S-VOC emissions and their impact on the atmospheric organic budget as well as the OA oxidation/oxygenation levels during different seasons is challenging in spite of the recent advances concerning the description of I/S-VOC (Stockwell et al., 2015; Ciarelli et al., 2017b). This work aims at evaluating how commonly used parameterizations and assumptions of I/S-VOC emissions and ageing perform to model the OA concentrations and properties in the western Mediterranean region in winter. To that end, the CTM from the air quality platform Polyphemus is used with different parameterizations of I/S-VOC emissions and ageing.

This paper is structured as follow: section 2 presents the setup of the air-quality model used and reference measurements. Section 3 presents the different emissions and ageing mechanisms used to describe the evolution of I/S-VOC as well as the comparison method. Section 4 compares the simulated concentrations, compositions of OA for the simulations using the different parameterizations. Finally, section 5 compares the measured and simulated OM:OC and O:C ratios.

## 2 Model and measurement set-up

The period of interest of this study is January-March 2014, hereafter referred to as the winter 2014 campaign.





## 2.1 General model setup

The Polyphemus/Polair3d air-quality model is used, with a similar setup as Chrit et al. (2017).

Transport and both dry and wet deposition are modeled folowing Sartelet et al. (2007). The Carbon Bond 05 model is used for gas-phase chemistry. Semi-volatile organic compounds formation mechanisms from five SOA gaseous precursors namely

isoprene, monoternenes, sesquiterpenes, aromatic compounds and intermediate and semi-volatile organic compounds from anthropogenic emissions (Kim et al., 2011; Couvidat et al., 2012) are added to CB05 model. Theses five precursors are modeled with a few surrogates as proxies to represent all the species. The aerosol dynamics (coagulation and condensation/evaporation) are modeled using the SIze REsolved Aerosol Model (SIREAM) (Debry et al., 2007) based on a sectional approach with an aerosol distribution of 24 sections of bound diameters: 0.01, 0.0141, 0.0199, 0.0281, 0.0398, 0.0562, 0.0794, 0.1121, 0.1585,

0.199, 0.25, 0.316, 0.4, 0.5, 0.63, 0.79, 1.0, 1.2589, 1.5849, 1.9953, 2.5119, 3.5481, 5.0119, 7.0795 and 10.0 $\mu$m.

The thermodynamic model used for condensation/evaporation of inorganic aerosol is ISORROPIA (Nenes et al., 1998) and the gas/particle partioning of SOA is computed with SOAP (Couvidat and Sartelet, 2015). In order to compute the gas/particle partitioning of both inorganics and inorganics, a bulk equilibrium approach is adopted. After condensation/evaporation, the mass is redistributed among size bins using the moving diameter algorithm.

The simulations are run between 01 January and 01 April 2014 for both the nesting (Europe) and the nested (Mediterranean) domains. The simulation domains (Europe and Mediterranean) and the spatial resolution used in the present study are the same as the ones used in Chrit et al. (2017). Boundary conditions for the European domain are obtained from the global chemistry-transport model MOZART v4.0 (Horowitz et al., 2003) (https://www.acom.ucar.edu/wrf-chem/mozart.shtml). The European simulation provides initial and boundary conditions to the Mediterranean one.

The European Center for Medium-Range Weather Forecasts (ECMWF) model provides the meteorological fields. The Troen and Mahrt parameterization (Troen and Mahrt, 1986) is used to compute the vertical diffusion. The land cover is modeled using the Global Land Cover 2000 (GLC-2000; http://www.gvm.jrc.it/glc2000/) data set. Sea-salt emissions are parameterized following Jaegle et al. (2011) and are assumed to be composed of sodium, chloride and sulfate (Schwier et al., 2015). Other sea-salt compounds, such as calcium, are not modelled. Biogenic emissions are estimated with the Model of Emissions of Gases

and Aerosols from Nature (MEGAN, Guenther et al. (2006)). Anthropogenic emissions are generated using the EDGAR-HTAP_V2 inventory for 2010 (http://edgar.jrc.ec.europa.eu/htap_v2/). The monthly and daily temporal distribution for the different activity sectors are obtained from GENEMIS (1994), and the hourly temporal distribution from Sartelet et al. (2012). $NO_x$, $SO_x$ and $PM_{2.5}$ emissions are speciated as described in Chrit et al. (2017). I/S-VOC gas-phase emissions are estimated from the POA emissions from residential heating by multiplying them by a constant factor assumed to be 1.5 in the default

simulation. As described in section 3.5, different values will be used and compared for I/S-VOC gas-phase emissions from residential heating and from other sectors. The I/S-VOC emissions from residential heating are assumed to be those of the sector "htap_6_residential" of the EDGAR-HTAP_V2 inventory. The emissions from this sector (shown in Figure 1) concern the emissions from heating/cooling and equipment/lightening of buildings as well as waste treatment. The I/S-VOC emissions from residential heating are obtained from the POA emissions of sector 6 by multiplying them by a constant factor noted $R_{RH}$





= I/S-VOC/POA. These emissions over the Mediterranean domain are located over big cities (Marseille, Milan, Rome, etc).
I/S-VOC emissions from the six other anthropogenic sources (shown in Figure 1) are estimated from the POA emissions by
multiplying them by a constant factor noted R = I/S-VOC/POA. These emissions are located over big cities and along the main
traffic routes, as well as the shipping routes linking Marseille to Ajaccio and Bastia. Different approaches will also be used to
represent the ageing of I/S-VOC, as described in section 3.

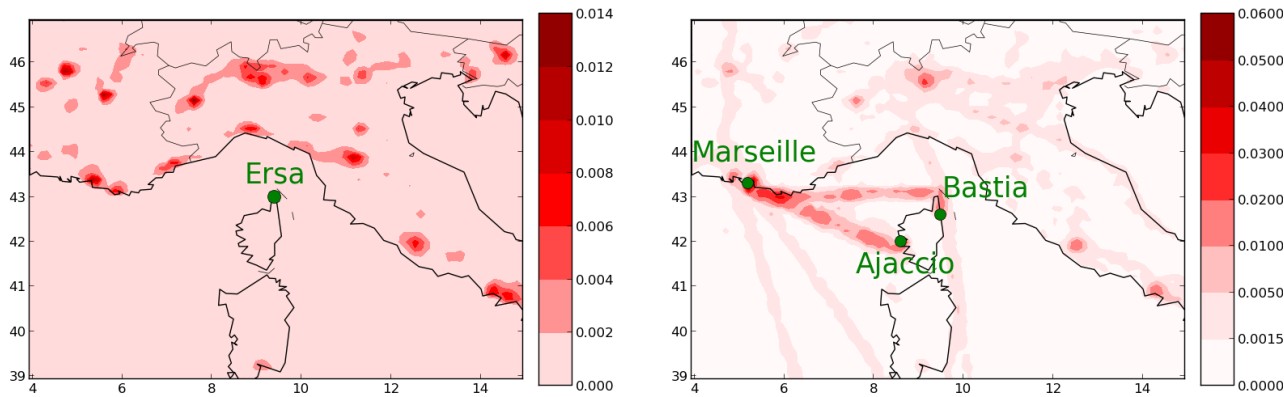

**Figure 1.** Surface emissions of POA from the residential heating sector (left panel) and from the other six anthropogenic sectors (right panel)
during the winter 2014. The emissions are in $\mu g.m^{-2}.s^{-1}$

## 2.2   Measurement setup

The ground-based measurements were performed in the framework of ChArMEx (The Chemistry-Aerosol Mediterranean
Experiment) at Ersa (42°58'N, 9°21.8'E) on a ridge at the northern tip of Corsica Island at an altitude of about 530 m.a.s.l..
The ground-based comparisons are performed by comparing the measured and modeled concentrations at the model cell the
closest to the station (42°52N, 9°22'30"E, 494 m.a.s.l.), as detailed in Chrit et al. (2017). An Aerodyne$^{TM}$ ACSM was used
in order to measure the near real-time mass concetration and chemical composition of aerosols with aerodynamic diameters
between 70 and 1000 nm with a time resolution of 30-min (Ng et al., 2011). This instrument has been continuously running
at Ersa between June 2012 and July 2014 (Nicolas, 2013), with an on-site set-up similar to the one presented in Michoud
et al. (2017). A recent intercomparison exercise, which he ACSM used in this study has successfully taken part in, report an
expanded uncertainty of 19% for OM (Crenn et al., 2015). OM:OC and O:C ratios are estimated using these measurements
following the methodology provided in Kroll et al. (2011). Although Crenn et al. (2015) and Fröhlich et al. (2015) have shown
consistent results (eg satisfactorily Z-scores) in terms of fragmentation pattern, higher discrepancies were observed for $f_{44}$
(mass fraction of m/z44), which is an essential variable in the calculation of these elemental ratios. In this respect, results are
presented with an uncertainty which can be estimated as being twice the one of PM (i.e. around 40%).



## 2.3 Model/measurements comparison method

To evaluate the performance of the model, we compare model simulation results to measurements at the Ersa site using a variety of performance statistical indicators. These indicators are: the simulated mean ($\bar{s}$), the root mean square error (RMSE), the correlation coefficient (corr), the mean fractional bias (MFB) and the mean fractional error (MFE). Table A1 of Appendix A lists the key statistical indicators definitions used in the model-to-data intercomparison. Furthermore, the criteria of Boylan and Russell (2006) (detailed in Table A2 of Appendix A) is used to assess the performance of the simulations.

## 3 Modeling of I/S-VOC emissions and ageing

In order to understand the behavior of the different parameterizations commonly used in CTMs to represent emissions and ageing of I/S-VOC in the western Mediterranean region, several simulations using different parameterizations are compared. These parameterizations are those described in Couvidat et al. (2012), Koo et al. (2014) and Ciarelli et al. (2017b). The differences concern the emission ratios used to estimate I/S-VOC from POA (R and $R_{RH}$), the ageing scheme (one step or multi-generational), the modeling of NTVOC, as well as the ratio OM:OC and volatility distribution at emissions.

### 3.1 One-step oxidation scheme

The one-step oxidation mechanism of Couvidat et al. (2012) is based on the fitting of the curve of dilution of POA from diesel exhaust of Robinson et al. (2007). I/S-VOC are modeled with three surrogate species POAlP, POAmP and POAhP of different volatilities characterized by their saturation concentrations (0.91, 86.21 and 3225.80 $\mu$g m$^{-3}$ respectively). The properties of the primary and aged I/S-VOC are shown in Table B1 of Appendix B. The ageing of each of these primary surrogates is modeled by a one-step OH-oxidation reaction in the gas phase (Appendix B), leading to the formation of secondary surrogates SOAlP, SOAmP and SOAhP. Once formed, these secondary surrogates do not undergo further oxidations. Compared to the primary surrogates, the volatility of the secondary surrogates is reduced by a factor of 100 and their molecular weight is increased by 40% (Grieshop et al., 2009; Couvidat et al., 2012) to represent functionalization and fragmentation.

### 3.2 Multi-generational step oxidation scheme

In sensitivity simulations, for anthropogenic I/S-VOC emissions, the oxidation mechanism is based on the hybrid volatility basis set (1.5-D VBS) approach developed by Koo et al. (2014). This mechanism combines the simplicity of the 1-dimensional (1-D) VBS with the ability to describe evolution of OA in the 2-dimensional space of oxidation state and volatility. This basis set uses five volatility surrogates, characterized by saturation concentrations varying between 0.1 and 1000 $\mu$g m$^{-3}$. The surrogates VAP0, VAP1, VAP2, VAP3 and VAP4 refer to the primary surrogates and VAS0, VAS1, VAS2, VAS3 and VAS4 refer to the secondary ones. Table C1 of Appendix C lists their properties.

In the scheme developed by Koo et al. (2014), the OH-oxidation of the primary surrogates leads to a mixture of primary and secondary surrogates of lower volatility. The carbon (oxygen respectively) number of the lower volatility surrogate decreases




(increases respectively) indicating that functionalization and fragmentation are implicitly accounted for. This mechanism is detailed in Appendix C.

## 3.3 Multi-generational step oxidation scheme for residential heating

In sensitivity simulations, for anthropogenic I/S-VOC emissions from residential heating, the VBS model developed by Ciarelli et al. (2017b) is also used. As in the previously detailed multi-step oxidation scheme, five surrogates with volatilities characterized by saturation concentrations extending from 0.1 to 1000 $\mu$g m$^{-3}$ are used. The primary surrogates (BBPOA1, BBPOA2, BBPOA3, BBPOA4, BBPOA5) react with OH to form secondary surrogates (BBSOA0, BBSOA1, BBSOA2, BB-SOA3, BBSOA4), whose volatility is one order of magnitude lower than the primary surrogate. In opposition to the one-step and multi-step oxidation schemes detailed above, here the secondary surrogates may also undergo OH-oxidation forming the secondary surrogate of lower volatility. As in the other schemes, functionalization and fragmentation are taken into account as the carbon and oxygen numbers of the secondary surrogates increases and decreases respectively. The properties of the VBS surrogates are shown in Table D1 of Appendix D, where reactions are also detailed.

Data from recent wood combustion and ageing experiments performed in smog chamber by Ciarelli et al. (2017b) show significant contribution of SOA from non-traditional volatile organic compounds (NTVOC: phenol, m-, o-, p-cresol, m-, o-, p-benzenediol/2-methylfuraldehyde, dimethylphenols, guaiacol/methylbenzenediols, naphthalene, 2-methylnaphthalene/1-methylnaphthalene, acenaphthylene, syringol, biphenyl/acenaphthene, dimethylnaphthalene) to OA mass. These NTVOC are usually not accounted as SOA precursors in CTMs. The NTVOC mixture saturation concentration is estimated to be $\sim$10$^6$ $\mu$g m$^{-3}$ falling with the IVOC saturation concentrations range limit (Koo et al., 2014; Donahue et al., 2012). NTVOC emissions are estimated using a ratio of NTVOC/SVOC of 4.75 (Ciarelli et al., 2017b) and their OH-oxidation produces four secondary surrogates of different volatilities. These four surrogates may undergo OH-oxidation leading to the less volatile and more oxidized secondary surrogate, similarly to the multi-step oxidation described in section 3.3. This mechanism is detailed in Appendix D and the surrogates properties are listed in Table D2 of Appendix D.

## 3.4 Volatility distribution and properties of primary emissions

In the one-step oxidation scheme of Couvidat et al. (2012), the emission distribution is based on the fitting of the curve of dilution of diesel exhaust from Robinson et al. (2007) and is shown in Table 1. This emission distribution is approximately similar to the one measured by May et al. (2013a) for biomass burning, and used in the multi-step oxidation scheme for residential heating of Ciarelli et al. (2017b). In the multi-step oxidation scheme of Koo et al. (2014) for anthropogenic emissions, the emission distribution is obtained from averaging the emission distributions from gasoline and diesel vehicles measured by May et al. (2013b, c). As shown in Table 1, the emitted I/S-VOC are less volatile than in the biomass-burning volatility distribution of May et al. (2013b). Here, the volatility distributions are assigned to a profile number (equal to 1 or 2), depending on whether the volatility profile is similar to the profile from biomass burning emissions of May et al. (2013b) (profile number



| Profil N° | | 1 | 2 | Profil N° | | 1 | 2 |
|---|---|---|---|---|---|---|---|
| Reference | | May et al. (2013b, c) | Couvidat et al. (2012) | Reference | | May et al. (2013b, c) | May et al. (2013a) |
| Saturation Conc. | 0.9 | 0.35 | 0.25 | Saturation Conc. | 0.1 | 0.15 | 0.20 |
| | | | | | 1 | 0.20 | 0.10 |
| | 86.2 | 0.51 | 0.32 | | 10 | 0.31 | 0.10 |
| | | | | | 100 | 0.20 | 0.20 |
| | 3225.8 | 0.14 | 0.43 | | 1000 | 0.14 | 0.4 |

**Table 1.** Summary of the volatility distributions of the primary I/S-VOC surrogates. Saturation concentrations are expresssed in $\mu$g m$^{-3}$.

| Profil N° | | 1 | 2 | Profil N° | | 1 | 2 |
|---|---|---|---|---|---|---|---|
| Reference | | Couvidat et al. (2012) | | Reference | | Koo et al. (2014) | Ciarelli et al. (2017b) |
| Saturation Conc. | 0.9 | 1.3 (0.15) | 1.7 (0.55) | Saturation Conc. | 0.1 | 1.36 (0.16) | 1.64 (0.37) |
| | | | | | 1 | 1.31 (0.12) | 1.53 (0.29) |
| | 86.2 | 1.3 (0.15) | 1.7 (0.55) | | 10 | 1.26 (0.07) | 1.44 (0.22) |
| | | | | | 100 | 1.21 (0.03) | 1.36 (0.15) |
| | 3225.8 | 1.3 (0.15) | 1.7 (0.55) | | 1000 | 1.17 (0) | 1.28 (0.09) |

**Table 2.** Summary of the OM:OC (and O:C) ratio of the primary I/S-VOC surrogates. Saturation concentrations are expresssed in $\mu$g m$^{-3}$.

2) or whether it is similar to the profile from vehicle emissions of May et al. (2013c) and May et al. (2013a) (profile number 1).

The one-step and multi-step oxidation schemes also differ in the OM:OC and O:C ratios of the emitted surrogates. In the one-step oxidation scheme of Couvidat et al. (2012), the OM:OC and O:C ratios are assumed to be constant (1.3) and close
5 to the average OM:OC and O:C ratios of Koo et al. (2014). However, for residential heating, the multi-oxidation scheme of Ciarelli et al. (2017b) assumes higher OM:OC and O:C rations, as described in Table 2. Here, the OM:OC and O:C ratios are assigned to a profile number (equal to 1 or 2), depending on whether the ratios are similar to the profile from biomass burning emissions of Ciarelli et al. (2017b) (profile number 2) or whether they are lower (profile number 1).

### 3.5 Sensitivity simulations

10 The setup of the different simulations is summarized in Table 3. The simulation S1 uses the setup commonly used in air-quality simulations with the Polyphemus platform: the one-step ageing scheme of Couvidat et al. (2012) is used for both residential heating and other anthropogenic sectors.



| Simulation | Residential heating | | | | | Other anthropogenic sectors | | | |
|---|---|---|---|---|---|---|---|---|---|
| | Ageing | Volatility profile | $R_{RH}$ | OM:OC profile | NTVOC | Ageing | Volatility profile | R | OM:OC profile |
| S1 | one-step (Couvidat) | 2 | 1.5 | 1 | No | one-step (Couvidat) | 2 | 1.5 | 1 |
| S2 | one-step (Couvidat) | 2 | 1.5 | 2 | No | one-step (Couvidat) | 1 | 1.5 | 1 |
| S3 | multi-step (Ciarelli) | 2 | 1.5 | 2 | No | multi-step (Koo) | 1 | 1.5 | 1 |
| S4 | multi-step (Ciarelli) | 2 | 1.5 | 2 | Yes | multi-step (Koo) | 1 | 1.5 | 1 |
| S5 | one-step (Couvidat) | 2 | 4.0 | 2 | No | one-step (Couvidat) | 1 | 1.5 | 1 |
| S6 | multi-step (Ciarelli) | 2 | 4.0 | 2 | Yes | multi-step (Koo) | 1 | 1.5 | 1 |

**Table 3.** Summary of the parameters used in the different simulations performed.

The simulation S2 is conducted to evaluate the impact of the volatility distribution of emissions. Instead of using a volatility distribution specific of biomass burning for all sectors as in S1, the volatility distribution specific of car emissions is used for anthropogenic sectors other than residential heating.

The simulation S3 is conducted to evaluate the impact of the ageing scheme. The volatility distributions are similar as S2, but multi-generational schemes are used rather than a single-oxidation strep for all anthropogenic sectors.

The simulation S4 is evaluated to estimate the impact of NTVOC. It has the same setup as S2 with multi-generational ageing, but NTVOC are taken into account.

The simulations S5 and S6 are conducted to assess the impact of the I/S-VOC/POA ratio used for residential heating ($R_{RH}$). The simulation S5 has the same setup as the simulation S2 (single-step oxidation), but it differs in the ratio $R_{RH}$, which is assumed to be equal to 4 rather than 1.5. The simulation S6 has the same setup as the simulation S4 (multi-step oxidation and NTVOC), but it differs in the ratio $R_{RH}$, which is assumed to be equal to 4 rather than 1.5.

In terms of the OM:OC ratio, the ratio specific of car emissions is used for emissions from anthropogenic sectors other than residential heating. For residential heating, higher OM:OC ratios are used in all simulations, except in S1, where the ratio specific of car emissions is used for all sectors.

## 4 Organic concentrations

The spatial distribution of $OM_1$ concentrations averaged over the first 3 months of 2014 (Figure E1 of Appendix E) shows that high $OM_1$ concentrations are mostly located over big cities like Marseille, Genoa, Turin, Milan, Rome and Naples and along maritime traffic routes, stressing that organics during wintertime are likely to be mostly of anthropogenic origins.

The simulated composition of $OM_1$ at Ersa is shown in Figure 2 for the simulations S4 and S5. In all simulations, primary and secondary organic aerosols (POA and SOA) from anthropogenic I/S-VOC are the main components of the organic mass (between 60% and 84%). POA tends to account for almost the same fraction of the organic mass than SOA (between 46% and 62%). Similarly, in the U.S., Koo et al. (2014) found that the SOA account for less than half of the modeled OA mass in winter





2005 due to the slow chemical ageing during the cold season. Over Europe, in March 2009, Ciarelli et al. (2017a) simulated that POA accounts between 12 and 68% of the OA, with an average value of 38%. The emission sector 6 (residential heating) has a large contribution to OA (between 31% and 33%). This is also in line with Ciarelli et al. (2017a) who found that over Europe in March 2009, the contribution of the residential sector to OA varies between 20% and 45% with an average value of

38%. Furthermore, this sector contributes more to SOA (between 42% and 52% of SOA from I/S-VOC) than to POA (between 17% and 31% of SOA from I/S-VOC), because their I/S-VOC emissions are more volatile. The contribution from aromatic VOC is low (lower than 3%), and when NTVOC are considered, they represent between 18% and 21% of the organic mass. The model simulations performed revealed that, for the winter of 2014, the biogenic OA fraction is low (15-18%). Ciarelli et al. (2017a) also estimated the biogenic contribution to the organic budget to be between 5 and 20% over Europe.

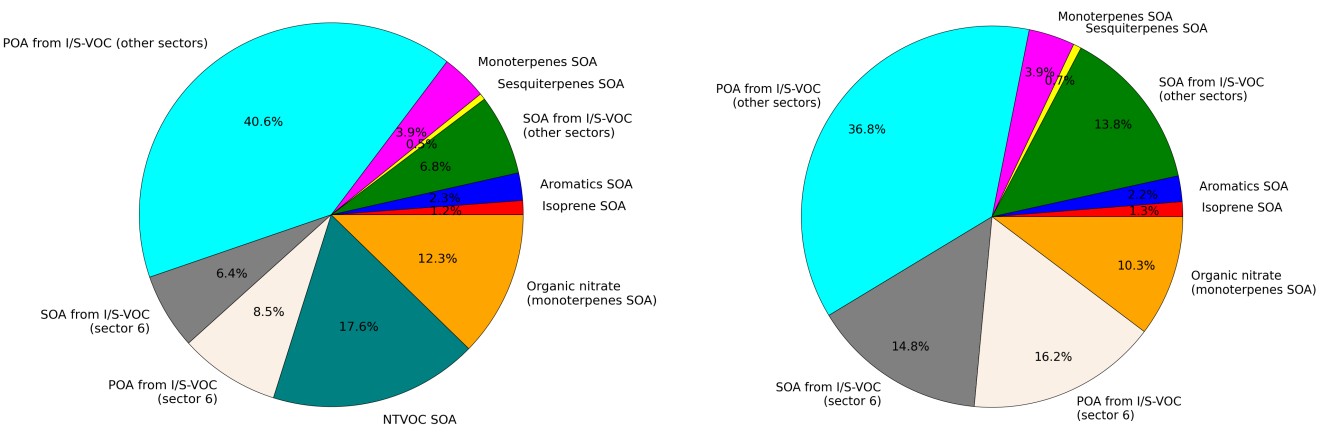

**Figure 2.** Simulated composition of $OM_1$ during the winter campaign of 2014 for two simulations: S4 (left panel) and S5 (right panel).

The statistical evaluation of the simulations is shown in Table 4. The performance criterion is satisfied for all simulations and the goal criterion is satisfied for S2, S3, S4 and S5. The goal criterion is not satisfied for the simulation S1, which uses single-step oxidation with a biomass-burning type volatility distribution for all anthropogenic sectors, and for the simulation S6, which uses multi-step oxidation with NTVOC and a high $R_{RH}$ ratio. The simulation S1 strongly under-estimates the $OM_1$ concentration at Ersa, whereas the simulation S6 strongly over-estimates it.

All the simulations tend to under-estimate the $OM_1$ concentrations at Ersa, except for the two simulations where NTVOC are taken into account (S4 and S6), which over-estimate the $OM_1$ concentrations at Ersa.

The model-to-measurement correlation is high for all simulations (between 76 and 83%).

Other CTMs showed the same under-estimation of $OM_1$ concentrations during winter over Europe, even when I/S-VOC emissions are taken into account (Couvidat et al., 2012; Denier van der Gon et al., 2015). The CTM CAMX (Comprehensive

Air Quality Model with extensions) also under-estimated the organic concentrations over Europe during February and March





| | Simulations | S1 | S2 | S3 | S4 | S5 | S6 |
|---|---|---|---|---|---|---|---|
| | $\overline{s} \pm$ RMSE | $0.75 \pm 1.14$ | $1.06 \pm 0.91$ | $1.20 \pm 0.85$ | $1.65 \pm 0.79$ | $1.25 \pm 0.80$ | $2.06 \pm 1.08$ |
| $\overline{o} = 1.45$ | Correlation (%) | 78.3 | 76.7 | 76.2 | 82.4 | 78.8 | 82.7 |
| | MFB (%) | -55 | -23 | -11 | 17 | -7 | 38 |
| | MFE (%) | 59 | 40 | 37 | 39 | 35 | 48 |

**Table 4.** Statistics of model to measurements comparisons for daily $OM_1$ concentrations during the winter campaign of 2014 at Ersa. $\overline{o}$ refers to the observed mean. Other statistical indicators are defined in Table A1 of AppendixA.

2009 (Ciarelli et al., 2017a), but considerable improvement was found for the modeled organic aerosol (OA) mass with the MFB decreasing from -61 to -29 %, when the parameterization of (Ciarelli et al., 2017b) with NTVOC was added.

The model-to-measurement comparison during the first 3-months of 2014 in terms of the daily concentrations of $OM_1$ at Ersa is shown in Figure 3.

5    Globally, the temporal variations of the simulated concentrations are well reproduced by the model. The simulation S1, which uses single-step oxidation with a biomass-burning type volatility distribution for all anthropogenic sectors, under-estimates the peaks. However, the peaks are well reproduced by the simulations S2, S3 and S5. The simulations S4 and S6, which take into account NTVOC over-estimate the peaks. All simulations under-estimate the beginning of the peak between 9 and 15 March, probably due to uncertainties in meteorology especially rain episodes, and changes in the origin of air masses.

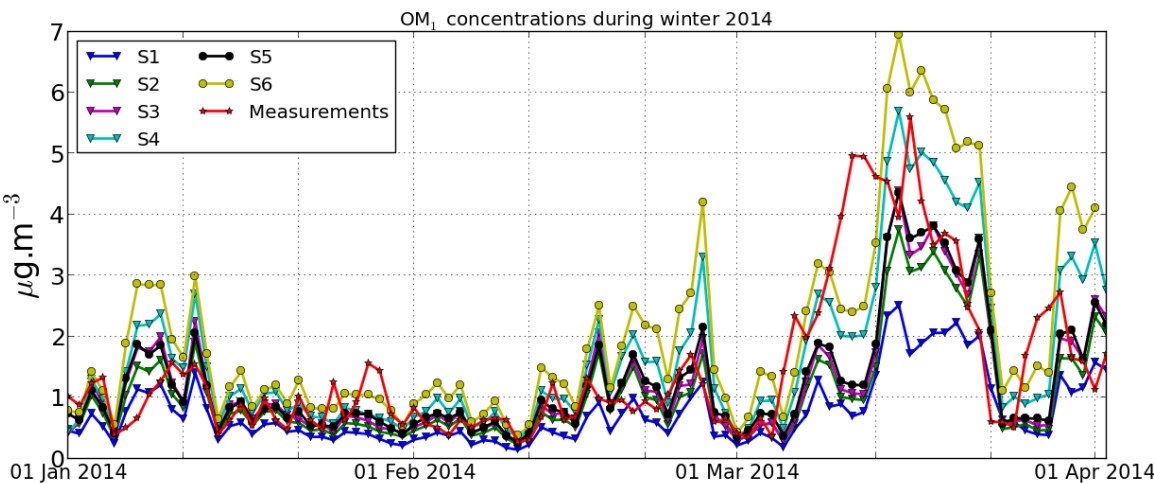

**Figure 3.** Daily evolution of measured and simulated $OM_1$ concentrations at Ersa from 1 January to 2 April.

10    As detailed in section 3.5, the difference between the simulations S2 and S1 originates in differences in the volatility distribution of emissions from anthropogenic sectors other than residential heating. In the simulation S2, a less volatile distribution is used than in the simulation S1, leading to larger OA concentrations in the particle phase. This difference in the volatility





distribution makes a large difference in the OA concentrations, removing the strong under-estimation simulated in simulation S1 (the MFB is -55% in S1 and only -23% in S2).

Considering multi-step ageing for all anthropogenic sectors also leads to an increase of OA concentrations (the MBF of the simulation S3 is -11%, which is lower in absolute value than the simulation S2). However, the influence of the multi-step ageing
(difference between S2 and S3 shown in Figure E1 of Appendix E) is lower than the influence of the volatility distribution (difference between S1 and S2 shown in Figure E1 of Appendix E). This larger influence of the volatility distribution than the multi-step ageing is true not only at Ersa, but also over the whole Mediterranean domain, where the average RMSE between the simulations S1 and S2 is 0.01 $\mu$g m$^{-3}$ (impact of volatility), against 0.005 $\mu$g m$^{-3}$ for the RMSE between the simulations S2 and S3 (impact of multi-step ageing).

At Ersa, increasing the ratio R$_{RH}$ from 1.5 to 4 (difference between simulation S3 and S2 shown in Figure E1 of Appendix E) has almost the same impact as considering the multi-step ageing (difference between simulations S5 and S2 shown in Figure E1 of Appendix E), although the statistics are slightly better when the ratio R$_{RH}$ is increased from 1.5 to 4 than when multi-step ageing is considered. However, this is not true over the whole Mediterranean domain, where the impact of increasing the ratio R$_{RH}$ from 1.5 to 4 is large over cities, whereas the impact of multi-step ageing stays low (see Figure E1 of Appendix E). Over
the whole Mediterranean domain, the average RMSE between the simulations S2 and S5 is 0.014 $\mu$g m$^{-3}$ (impact of increasing the ratio R$_{RH}$ from 1.5 to 4), against 0.005 $\mu$g m$^{-3}$ for the RMSE between the simulations S2 and S3 (impact of multi-step ageing).

Although considering NTVOC leads to a slight increase in correlation, it also leads to an over-estimation of OA concentrations at Ersa. Over the whole Mediterranean domain, the impact of NTVOC is high with an average RMSE between the
20 simulations S4 and S3 of 0.0211 $\mu$g m$^{-3}$.

Finally, the best statistics, in terms of MFE and MFB are obtained for the simulation S5, with a one-step ageing scheme, a volatility distribution typical of biomass burning for the residential sector with a ratio R$_{RH}$ of 4, and a volatility distribution typical of car emissions for other sectors with a ratio R of 1.5.

## 5 Oxidation and oxygenation of organics

The oxidation state is quantified using two metrics: OM:OC and O:C calculated as detailed in Chrit et al. (2017). Figure 4 shows the daily variations of the measured and simulated ratios for the different simulations.



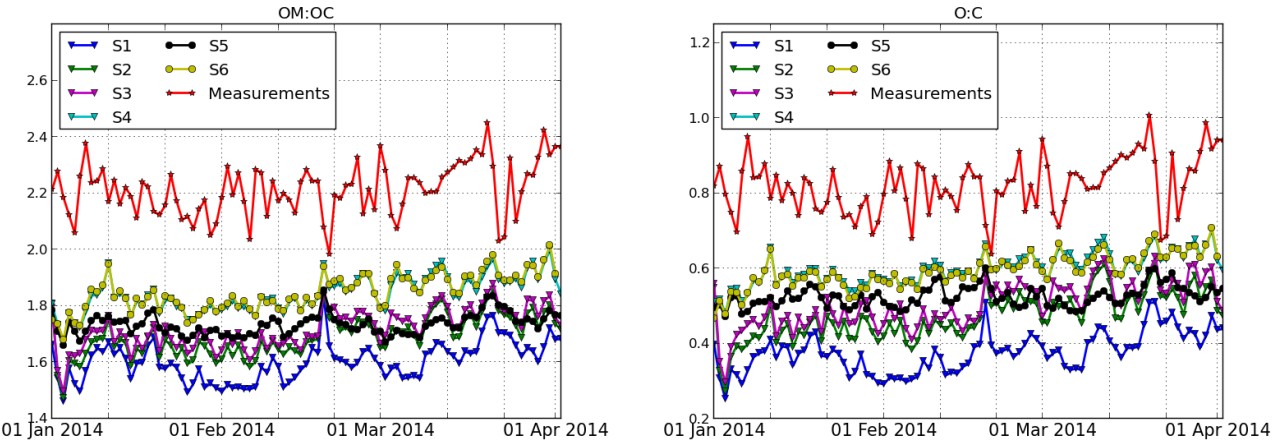

**Figure 4.** Daily evolution of the ratios OM:OC (left panel) and O:C (right panel) from 01 January to 02 April 2014 at Ersa.

The measurements at Ersa show highly oxidized and oxygenated organics: the measured OM:OC and O:C ratios at Ersa are respectively $2.21 \pm 0.09$ and $0.82 \pm 0.07$ These values are lower than the index measured during the summer 2013 by Chrit et al. (2017) ($2.43 \pm 0.07$ and $0.99 \pm 0.06$ for the measured OM:OC and O:C ratios at Ersa respectively), due to the slower oxidation process owing to the lower temperatures during winter. The average simulated OM:OC and O:C ratios are

5   shown in Table 5. Both index are strongly underestimated by all simulations, due to the high contribution of POA to the $OM_1$ concentrations (POA is less volatile and oxygenated than SOA). The simulations using multi-step ageing schemes for I/S-VOC emissions have higher OM:OC and O:C ratios, although the differences are very low (the OM:OC ratio is $1.69 \pm 0.53$ in S2 (single-step) and $1.72 \pm 0.50$ in S3 (multi-step). Organics in the simulations where the strength of I/S-VOC emission from residential heating was increased (simulations S5 and S6) have higher OM:OC and O:C ratios because POA and SOA from

10   I/S-VOC from residential heating are more oxidized and oxygenated than POA and SOA from other anthropogenic sources. Similarly, organics in the simulations where NTVOC are taken into account have higher OM:OC and O:C ratios, because in the model, NTVOC lead to very oxidized and oxygenated OA. However, the simulated ratios OM:OC and O:C stay underestimated ($1.85 \pm 0.38$ and $0.60 \pm 0.24$ at most, against $2.21 \pm 0.09$ and $0.82 \pm 0.07$ in the measurements).

| Simulations | S1 | S2 | S3 | S4 | S5 | S6 | Measurements |
|---|---|---|---|---|---|---|---|
| OM:OC | $1.60 \pm 0.62$ | $1.69 \pm 0.53$ | $1.72 \pm 0.50$ | $1.85 \pm 0.38$ | $1.74 \pm 0.49$ | $1.85 \pm 0.38$ | $2.21 \pm 0.09$ |
| O:C | $0.38 \pm 0.45$ | $0.47 \pm 0.36$ | $0.50 \pm 0.33$ | $0.60 \pm 0.23$ | $0.53 \pm 0.31$ | $0.59 \pm 0.24$ | $0.82 \pm 0.07$ |

**Table 5.** Daily averages of OM:OC and O:C ratios at Ersa during winter 2014 for different simulations. The average measured OM:OC ratio is 2.21 and the average measured O:C ratio is 0.82.



# 6 Conclusion

This study shows a ground-based comparison of both modeled organic concentrations and properties to measurements performed at Ersa (Cape Corsica, France) during the winter 2014. This work aims at evaluating how commonly used parameterizations and assumptions of intermediate/semi-volatile organic compound (I/S-VOC) emissions and ageing perform in modeling the organic aerosol (OA) concentrations and properties in the western Mediterranean region in winter. To that end, the chemistry-transport model from the air quality platform Polyphemus is used with different parameterizations of I/S-VOC emissions and ageing (different volatility distribution emissions, single-step oxidation vs multi-step oxidation within a Volatility Basis Set framework, including non-traditional volatile organic compounds NTVOC). Winter (JFM) 2014 simulations are performed and compared to measurements obtained with an ACSM at the background station of Ersa in the North of Corsica Island. In all simulations, OA at Ersa is mainly from anthropogenic sources (15 to 18% of OA is from biogenic sources). The emission sector 6 (residential heating) has a large contribution to OA (between 31 and 33%). The contribution from aromatic VOC is low (lower than 3%). NTVOC, as modeled with the parameterization of Ciarelli et al. (2017b) represent between 18% and 21% of the organic mass. For most simulations, the concentrations of OA compare well to the measurements. All the simulations tend to under-estimate the OA concentrations at Ersa, except for the two simulations where NTVOC are taken into account, which, however, over-estimate the OA concentrations. Over the whole western Mediterranean domain, the volatility distribution at the emission influences more strongly the concentrations than the choice of the parameterization that may be used for ageing (single-step oxidation vs multi-step oxidation). Modifying the volatility distribution of sectors other than residential heating leads to a decrease of 29% in OA concentrations at Ersa, while using the multi-step oxidation parameterization rather than the single-step one leads to an increase of 13%. The best statistics are obtained using two configurations: the first one is a one-step ageing scheme, a volatility distribution typical of biomass burning for the residential sector with a ratio I/S-VOC/POA at emission of 4, and the second one is a multi-generational ageing scheme, a volatility distribution typical of car emissions for other sectors with a ratio R I/S-VOC/POA at emission of 1.5.

Both the OM:OC and O:C ratios are underestimated at Ersa in all simulations. The largest simulated OM:OC ratio is equal to $1.85 \pm 0.83$, against $2.21 \pm 0.09$ in the measurements. For the summer campaign, Chrit et al. (2017) improved the simulated OM:OC ratio by adding the formation mechanisms of both extremely-low volatile organic compounds and organic nitrate from monoterpene oxidation. Similarly, the formation of organic nitrate and highly oxygenated organic molecules (Molteni et al., 2018) from aromatic precursors should be added in order to better reproduce the observed OA oxidation/oxygenation levels. However, adding these new OA formation pathways may lead to an increase in OA concentrations, suggesting that the actual parameterizations, particularly those with NTVOC may need to be revisited, for example by better characterizing their deposition.

*Acknowledgements.* This research was funded by the French National Research Agency (ANR) projects SAF-MED (grant ANR-12-BS06-0013). It is part of the ChArMEx project supported by ADEME, CNRS-INSU, CEA and Météo-France through the multidisciplinary programme MISTRALS (Mediterranean Integrated Studies aT Regional And Local Scales). It contributes to ChArMEx work packages 1 and 2





on emissions and aerosol ageing, respectively. The ACSM at Ersa was funded by the CORSiCA project funded by the Collectivité Territoriale de Corse through the Fonds Européen de Développement Régional of the European Operational Program 2007-2013 and the Contrat de Plan Etat-Région. Eric Hamounou is acknowledged for his great help in setting up the Ersa station. CEREA is a member of the Institut Pierre-Simon Laplace (IPSL).



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





## Appendix A: Statistical indicators and criteria

| Statistic indicator | Definition |
|---|---|
| Root mean square error (RMSE) | $\sqrt{\frac{1}{n}\sum_{i=1}^{n}(c_i - o_i)^2}$ |
| Correlation (Corr) | $\dfrac{\sum_{i=1}^{n}(c_i - \bar{c})(o_i - \bar{o})}{\sqrt{\sum_{i=1}^{n}(c_i - \bar{c})^2}\sqrt{\sum_{i=1}^{n}(o_i - \bar{o})^2}}$ |
| Mean fractional bias (MFB) | $\frac{1}{n}\sum_{i=1}^{n}\dfrac{c_i - o_i}{(c_i + o_i)/2}$ |
| Mean fractional error (MFE) | $\frac{1}{n}\sum_{i=1}^{n}\dfrac{\mid c_i - o_i \mid}{(c_i + o_i)/2}$ |

**Table A1.** Definitions of the statistics used in this work. $(o_i)_i$ and $(c_i)_i$ are the observed and the simulated concentrations at time and location i, respectively. $n$ is the number of data

| Criteria | Performance criterion | Goal criterion |
|---|---|---|
| |MFB| | $\leq 60\%$ | $\leq 30\%$ |
| MFE | $\leq 75\%$ | $\leq 50\%$ |

**Table A2.** Boylan and Russel criteria





## Appendix B: Single-step ageing of I/S-VOC (Couvidat et al., 2012)

$$POAlP + OH \xrightarrow{k} SOAlP \tag{B1}$$

$$POAmP + OH \xrightarrow{k} SOAmP \tag{B2}$$

$$POAhP + OH \xrightarrow{k} SOAhP \tag{B3}$$

5 $$\tag{B4}$$

with k = $2.0 \times 10^{-11}$ cm$^3$. molecule$^{-1}$. s$^{-1}$.

| Surrogate | Emission fraction | Molecular weight | $\log_{10}(C^*)$ at 298K | $\Delta H_{vap}$ | OM/OC | O/C |
|---|---|---|---|---|---|---|
| POAlP | 0.25 | 280 | -0.04 | 106.0 | 1.3 | 0.15 |
| POAmP | 0.32 | 280 | 1.94 | 91.0 | 1.3 | 0.15 |
| POAhP | 0.43 | 280 | 3.51 | 79.0 | 1.3 | 0.15 |
| SOAlP | — | 392 | -2.04 | 106.0 | 1.82 | 0.56 |
| SOAmP | — | 392 | -0.06 | 91.0 | 1.82 | 0.56 |
| SOAhP | — | 392 | 1.51 | 79.0 | 1.82 | 0.56 |

**Table B1.** Properties of the primary and secondary anthropogenic I/S-VOC. The molecular weights are in g.mol$^{-1}$. $\Delta H_{vap}$ is the enthalpy of vaporisation in KJ.mol$^{-1}$, which describes the temperature dependance of the saturation pressure C$^*$.

## Appendix C: Multi-step ageing of I/S-VOC (Koo et al., 2014)

$$VAP1 + OH \xrightarrow{k} 0.864VAP0 + 0.142VAS0 \tag{C1}$$

$$VAP2 + OH \xrightarrow{k} 0.877VAP1 + 0.129VAS1 \tag{C2}$$

10 $$VAP3 + OH \xrightarrow{k} 0.889VAP2 + 0.116VAS2 \tag{C3}$$

$$VAP4 + OH \xrightarrow{k} 0.869VAP3 + 0.137VAS3 \tag{C4}$$

$$\tag{C5}$$

with k = $4.0 \times 10^{-11}$ cm$^3$. molecule$^{-1}$. s$^{-1}$.




| Surrogate | Emission fraction | Molecular weight | $\log_{10}(C^*)$ at 298K | $\Delta H_{vap}$ | OM/OC | O/C |
|---|---|---|---|---|---|---|
| VAP0 | 0.15 | 278 | -1 | 96.0 | 1.36 | 0.16 |
| VAP1 | 0.20 | 275 | 0 | 85.0 | 1.31 | 0.12 |
| VAP2 | 0.31 | 272 | 1 | 74.0 | 1.26 | 0.07 |
| VAP3 | 0.20 | 268 | 2 | 63.0 | 1.21 | 0.03 |
| VAP4 | 0.14 | 266 | 3 | 55.0 | 1.17 | 0.00 |
| VAS0 | — | 172 | -1 | 35 | 2.05 | 0.70 |
| VAS1 | — | 167 | 0 | 35 | 1.92 | 0.60 |
| VAS2 | — | 163 | 1 | 35 | 1.81 | 0.51 |
| VAS3 | — | 158 | 2 | 35 | 1.70 | 0.43 |
| VAS4 | — | 153 | 3 | 35 | 1.59 | 0.34 |

**Table C1.** Properties of the VBS species (the primary and secondary anthropogenic SVOC). The molecular weights are in g.mol$^{-1}$. $\Delta H_{vap}$ is the enthalpy of vaporisation in KJ.mol$^{-1}$, which describes the temperature dependance of the saturation pressure $C^*$.

## Appendix D: Multi-step ageing of I/S-VOC from residential heating (Ciarelli et al., 2017b)

$$BBPOA4 + OH \xrightarrow{k} BBSOA3 \tag{D1}$$

$$BBPOA3 + OH \xrightarrow{k} BBSOA2 \tag{D2}$$

$$BBPOA2 + OH \xrightarrow{k} BBSOA1 \tag{D3}$$

$$BBPOA1 + OH \xrightarrow{k} BBSOA0 \tag{D4}$$

$$BBSOA3 + OH \xrightarrow{k} BBSOA2 \tag{D5}$$

$$BBSOA2 + OH \xrightarrow{k} BBSOA1 \tag{D6}$$

$$BBSOA1 + OH \xrightarrow{k} BBSOA0 \tag{D7}$$

$$NTVOC + OH \xrightarrow{k} 0.143 BB3SOA4 + 0.097 BB3SOA3 + 0.069 BB3SOA2 + 0.011 BB3SOA1 \tag{D8}$$

$$BB3SOA4 + OH \xrightarrow{k} BB3SOA3 \tag{D9}$$

$$BB3SOA3 + OH \xrightarrow{k} BB3SOA2 \tag{D10}$$

$$BB3SOA2 + OH \xrightarrow{k} BB3SOA1 \tag{D11}$$

$$BB3SOA1 + OH \xrightarrow{k} BB3SOA0 \tag{D12}$$

$$\tag{D13}$$

with k = $4.0 \times 10^{-11}$ cm$^3$. molecule$^{-1}$. s$^{-1}$.





| Surrogate | Emission fraction | Molecular weight | $\log_{10}(C^*)$ at 298K | $\Delta H_{vap}$ | OM/OC | O/C |
|-----------|-------------------|------------------|--------------------------|------------------|-------|-----|
| NTVOC   | 4.75 | 113 | 6  | —    | —    | —    |
| BBPOA0  | 0.20 | 216 | -1 | 85.0 | 1.64 | 0.37 |
| BBPOA1  | 0.10 | 216 | 0  | 77.5 | 1.53 | 0.29 |
| BBPOA2  | 0.10 | 216 | 1  | 70.0 | 1.44 | 0.22 |
| BBPOA3  | 0.20 | 216 | 2  | 62.5 | 1.36 | 0.15 |
| BBPOA4  | 0.40 | 215 | 3  | 55.0 | 1.28 | 0.09 |
| BBSOA0  | —    | 194 | -1 | 35.0 | 1.80 | 0.50 |
| BBSOA1  | —    | 189 | 0  | 35.0 | 1.70 | 0.43 |
| BBSOA2  | —    | 184 | 1  | 35.0 | 1.61 | 0.36 |
| BBSOA3  | —    | 179 | 2  | 35.0 | 1.53 | 0.29 |
| BB3SOA0 | —    | 149 | -1 | 35.0 | 2.48 | 1.05 |
| BB3SOA1 | —    | 144 | 0  | 35.0 | 2.29 | 0.90 |
| BB3SOA2 | —    | 140 | 1  | 35.0 | 2.12 | 0.76 |
| BB3SOA3 | —    | 135 | 2  | 35.0 | 1.96 | 0.63 |
| BB3SOA4 | —    | 131 | 3  | 35.0 | 1.82 | 0.52 |

**Table D1.** Properties of the VBS species (the NTVOC and primary and secondary RH-I/S-VOC). The molecular weights are in g.mol$^{-1}$. $\Delta H_{vap}$ is the enthalpy of vaporisation in KJ.mol$^{-1}$, which describes the temperature dependance of the saturation pressure C$^*$.





| Surrogate | Emission fraction | Molecular weight | $\log_{10}(C^*)$ at 298K | $\Delta H_{vap}$ | OM/OC | O/C |
|-----------|-------------------|------------------|--------------------------|------------------|-------|-----|
| NTVOC | 4.75 | 113 | 6 | — | — | — |
| BBPOA0 | 0.20 | 216 | -1 | 85.0 | 1.64 | 0.37 |
| BBPOA1 | 0.10 | 216 | 0 | 77.5 | 1.53 | 0.29 |
| BBPOA2 | 0.10 | 216 | 1 | 70.0 | 1.44 | 0.22 |
| BBPOA3 | 0.20 | 216 | 2 | 62.5 | 1.36 | 0.15 |
| BBPOA4 | 0.40 | 215 | 3 | 55.0 | 1.28 | 0.09 |
| BBSOA0 | — | 194 | -1 | 35.0 | 1.80 | 0.50 |
| BBSOA1 | — | 189 | 0 | 35.0 | 1.70 | 0.43 |
| BBSOA2 | — | 184 | 1 | 35.0 | 1.61 | 0.36 |
| BBSOA3 | — | 179 | 2 | 35.0 | 1.53 | 0.29 |
| BB3SOA0 | — | 149 | -1 | 35.0 | 2.48 | 1.05 |
| BB3SOA1 | — | 144 | 0 | 35.0 | 2.29 | 0.90 |
| BB3SOA2 | — | 140 | 1 | 35.0 | 2.12 | 0.76 |
| BB3SOA3 | — | 135 | 2 | 35.0 | 1.96 | 0.63 |
| BB3SOA4 | — | 131 | 3 | 35.0 | 1.82 | 0.52 |

**Table D2.** Properties of the NTVOC and their oxidation products). The molecular weights are in g.mol$^{-1}$. $\Delta H_{vap}$ is the enthalpy of vaporisation in KJ.mol$^{-1}$, which describes the temperature dependance of the saturation pressure C$^*$.





# Appendix E: Maps of OM$_1$ concentrations and differences between simulations.

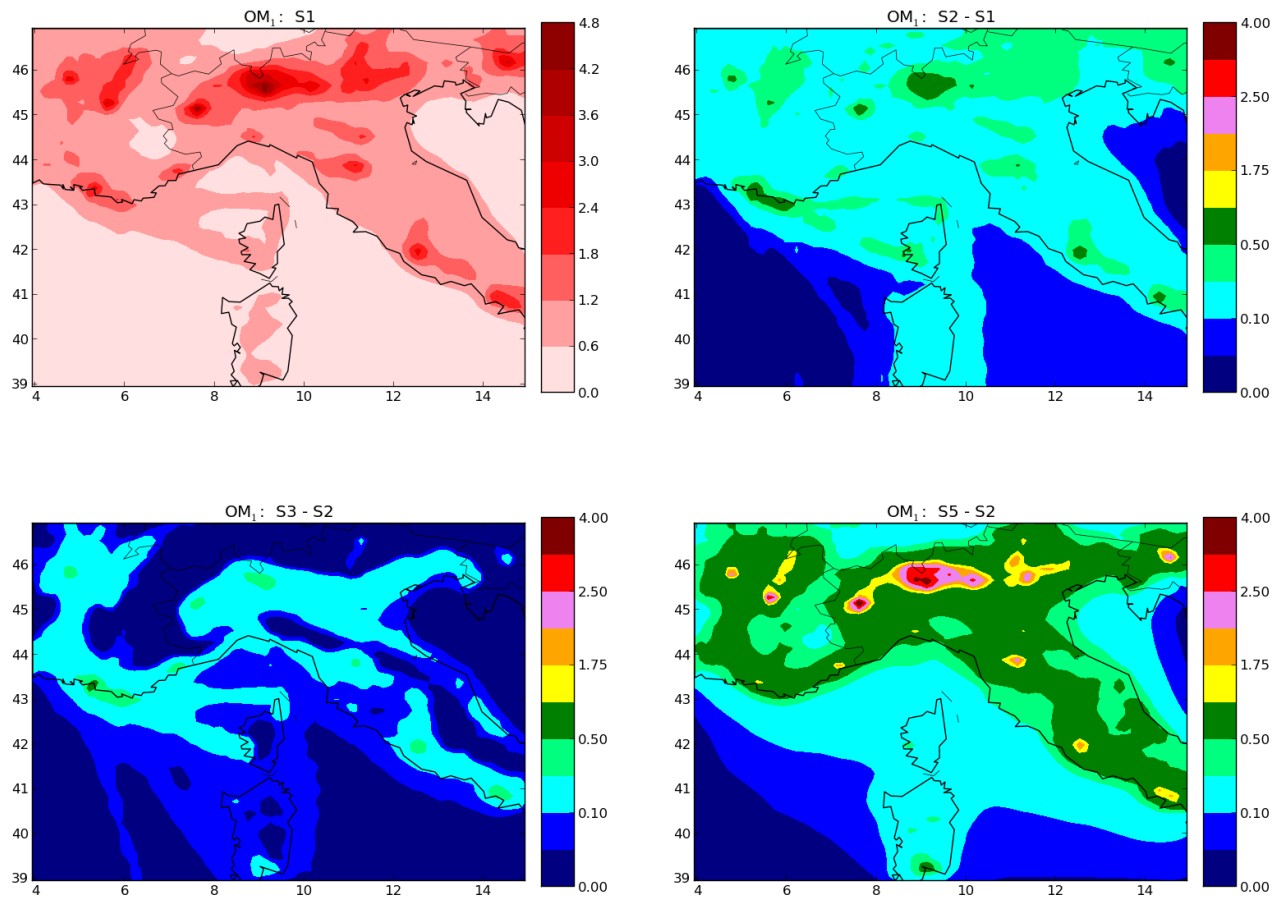

**Figure E1.** Maps of the concentrations of OM$_1$ ($\mu$g m$^{-3}$) averaged over January to March 2014 using S1 (upper left panel) and the absolute difference of OM$_1$ concentrations between S2 and S1 (upper right panel, impact of volatility), S3 and S2 (lower left panel, impact of multi-step ageing), and S5 and S2 (lower right panel, impact of increasing R$_{RH}$ from 1.5 to 4.