# Peer review of "Modeling organic aerosol concentrations and properties during winter 2014 in the northwestern Mediterranean region"

_Atmospheric Chemistry and Physics, 2018_

## Referee Comment (RC1) · Anonymous Referee #1 · 6 Aug 2018

In this work, Chrit et al. used the air quality model Polyphemus to describe the organic aerosol formation and properties (notably oxidation state) at a measurement site in Corsica during the winter campaign of 2014. The OA concentrations are well simulated by the model, however, their oxidation state is systematically underestimated compared to observations. They also stress the importance of an accurate characterization of emissions since they found that the volatility distribution at emissions is the prime factor that control the simulated OA concentration levels. Overall, the manuscript is well written and scientifically sound. I recommend this study for publication after taking the following comments into account.

**Specific comments:**

1. Page 1, lines 15-16: What is the difference between oxidation and oxygenation state of OA? If there is no difference please remove the term "oxygenation state" from this sentence

2. Page 1, lines 16-17: Why do you assume that only the multigenerational ageing of the residential heating OA can improve substantially the results? Is this the dominant sector in the area? What about the multigenerational ageing of OA from other sources?

3. Page 2, line 7: Please replace OA with POA.

4. Page 2, line 3: You can also add the work of Jathar et al. (2014) and Tsimpidi et al. (2017).

5. Page 2, line 30: You can also add the work of van der Gon et al. (2015)

6. Page 4, line 4: What is the difference between highly oxidized and highly oxygenated OA?

7. Page 4, line 8: These studies are not recent. Please add more recent studies, e.g. (Aiken et al., 2008; Tost and Pringle, 2012; Canagaratna et al., 2015; Tsimpidi et al., 2018)

8. Page 5, section 2.1: OA formation from alkanes, olefins, S/I-VOC from open biomass burning, and marine OA is missing from the model setup. Can you please add a comment on their potential importance in the examined area?

9. Page 5, line 13: Please change „inorganics and inorgancs" with „inorganics and organics".

10. Page 5, line 14: Please add a reference for the algorithm.

11. Page 5, line 15: According to the presented results (e.g., Fig. 3), the simulation lasts until $2^{nd}$ of April.

12. Page 5, line 16: It would be convenient if you can state the spatial resolution used here.

13. Page 5, lines 23-24: Please remove the sentence: "Other sea-salt…. not modelled"

14. Page 5 line 34: How much is this constant factor ($R_{RH}$)? And how much is the constant factor R stated later in the text?

15. Page 6, section 2.2: PMF analysis results would be very useful for comparison with your model results. Are they available at the Ersa site? If so, please add this comparison.

16. Page 6, line 10: Are these coordinated the center of the model cell? Does the dimension of the model cell include the coordinates of the station mentioned above?

17. Page 6, line 10: Please define the abbreviation ACSM

18. Page 6, line 12: Did you compare the measurements with the model results with size sections from 0.056 to 1.0 (as they appear in the previous page)? Please clarify.

19. Page 6, line 14: Please correct "he" with "the"

20. Page 7, section 3: What about traditional VOCs? Are they subject to photochemical ageing?

21. Page 8, section 3.3: How do you treat OA from sources other than residential heating in this case? Do they follow the oxidation scheme described in section 3.2? In that case, can you justify why you use a different oxidation scheme especially for residential heating and not for other sources?

22. Page 8, line 11: The carbon number should decrease and oxygen number increases, please correct.

23. Page 8, Section 3.4:Can you provide the actual emission rates (e.g., in Tn yr$^{-1}$) of your OA precursor emissions (i.e., VOCs, NTVOCs, I/S – VOCs from different types of sources)?

24. Page 9, Tables 1 and 2: Please improve the quality of the tables. For example, you should include names of surrogate species that you assign these numbers, names of sensitivity tests, and what these numbers express (i.e., emission factors, O:C, OM/OC should not be stated only in the caption but also inside the tables).

25. Page 10, line 11: This is not very clear. You apply a factor of 4 in the initial emission inventory, and then, on top of that you apply a factor of 4.75 to account for the NTVOC (which are not part of your S/I-VOC). Can you please clarify and justify your hypothesis of such high additional emissions?

26. Page 10, line 17: Please add in the sentence the average OM concentrations over these cities. Likewise, provide average concentrations for other mentioned areas (e.g., Ersa) later throughout the text.

27. Page 10, line 19: Why do you focus only in these two simulations?

28. Page 10, line 19: Do you mean in both simulations (instead of "in all simulations")?

29. Page 11, Figure 2: Please increase the font size of the fractions. Also, the fractions over the dark blue are not clear.

30. Page 12, line 1: OA is already defined.

31. Is this an assumption or did you actually check that you have a false rain episode in your model?

32. Page 15, line 5: OA is already defined.

33. Page 15, Section 6: Can you comment on the importance of marine OA in your domain? Is this type of OA identified by measurements at Ersa site?

34. Appendix A, table A1: The definitions are not clearly readable.

Aiken, A. C., Decarlo, P. F., Kroll, J. H., Worsnop, D. R., Huffman, J. A., Docherty, K. S., Ulbrich, I. M., Mohr, C., Kimmel, J. R., Sueper, D., Sun, Y., Zhang, Q., Trimborn, A., Northway, M., Ziemann, P. J., Canagaratna, M. R., Onasch, T. B., Alfarra, M. R., Prevot, A. S. H., Dommen, J., Duplissy, J., Metzger, A., Baltensperger, U., and Jimenez, J. L.: O/C and OM/OC ratios of primary, secondary, and ambient organic aerosols with high-resolution time-of-flight aerosol mass spectrometry, Environmen. Sci. & Technol., 42, 4478-4485, 2008.

Canagaratna, M. R., Jimenez, J. L., Kroll, J. H., Chen, Q., Kessler, S. H., Massoli, P., Ruiz, L. H., Fortner, E., Williams, L. R., Wilson, K. R., Surratt, J. D., Donahue, N. M., Jayne, J. T., and Worsnop, D. R.: Elemental ratio measurements of organic compounds using aerosol mass spectrometry: characterization, improved calibration, and implications, Atmospheric Chemistry and Physics, 15, 253-272, 2015.

Jathar, S. H., Gordon, T. D., Hennigan, C. J., Pye, H. O. T., Pouliot, G., Adams, P. J., Donahue, N. M., and Robinson, A. L.: Unspeciated organic emissions from combustion sources and their influence on the secondary organic aerosol budget in the United States, Proceedings of the National Academy of Sciences of the United States of America, 111, 10473-10478, 2014.

Tost, H. and Pringle, K. J.: Improvements of organic aerosol representations and their effects in large-scale atmospheric models, Atmospheric Chemistry and Physics, 12, 8687-8709, 2012.

Tsimpidi, A. P., Karydis, V. A., Pandis, S. N., and Lelieveld, J.: Global-scale combustion sources of organic aerosols: sensitivity to formation and removal mechanisms, Atmospheric Chemistry and Physics, 17, 7345-7364, 2017.

Tsimpidi, A. P., Karydis, V. A., Pozzer, A., Pandis, S. N., and Lelieveld, J.: ORACLE 2-D (v2.0): An efficient module to compute the volatility and oxygen content of organic aerosol with a global chemistry - climate model, Geosci. Model Dev. Discuss., 2018, 1-45, 2018.

van der Gon, H., Bergstrom, R., Fountoukis, C., Johansson, C., Pandis, S. N., Simpson, D., and Visschedijk, A. J. H.: Particulate emissions from residential wood combustion in Europe revised estimates and an evaluation, Atmospheric Chemistry and Physics, 15, 6503-6519, 2015.

---

## Referee Comment (RC2) · Anonymous Referee #2 · 9 Aug 2018

Comment on "Modeling organic aerosol concentrations and properties during winter 2014 in the northwestern Mediterranean region" by Chrit et al.

acp-2018-149

**General Comments:**

Chrit et al. have deployed the Polyphemus platform with expanded techniques for simulating organic aerosol formation and aging, particularly from residential heating sources, and applied it to the CharMEx campaign in winter 2014. The study design is generally sound and the sensitivity choices are informative. There is also adequate reporting of direct results. However, I found there to be a lack of further diagnosis or interpretation of results considering the amount of data the authors would have access to from the model. Considering this, I think the paper would be better suited in its current form for GMD, although I think it is of acceptable quality and scope for publication in ACP, given that the authors address specific concerns below.

**Specific Comments**

1. Section 3.4: I did not understand how profiles 1 and 2 are allocated to individual emissions. It should be made clearer why profiles 1 and 2 can be mismatched for volatility and O:C? Usually, I think of these as tied together by the molecular weight (i.e. carbon number) assumed for each model species of a given volatility. Are the assignments made using some feature from the emissions inputs, or are individual sectors assigned profile 1 or 2 based on some knowledge of their emissions (e.g. waste burning goes with profile 2, offroad diesel goes with profile number 1, etc)? If the latter, can the authors include a table that identifies these assignments? If the former, can the authors better describe what parameters and algorithm are used to make the assignments?
2. After going back to Ciarelli et al. (2017b), I am not convinced they included additional IVOCs, consistent with those being added in this simulation, in their parameterizations. It seems from Table 1 in that paper, that the authors included NTVOCs and also evaporated the existing POA into SVOC and IVOC bins. But I do not think they considered an additional IVOC category. Given this, I am not surprised that simulations here which include additional IVOCs and NTVOCs (S4 and S6) tend to overpredict measurements. I would suggest the authors perform at least one run with R_RH set to 0 for residential heating sources and NTVOC turned on. This will probably look a lot like S5 so if the authors want to adjust the explanation of their cases to avoid doing more simulations, I think that is okay, but some detailed explanation should be added (i.e. R_RH could be defined as adding NTVOC). Note this approach would not be perfect, because the SOA yields for the IVOCs would differ from those Ciarelli et al. (2017b) derived for NTVOC.
3. Are the IVOCs from residential heating assumed to be the same composition (i.e. same SOA yields) as those from vehicle sources? If so, what is this based on?
4. Do the authors have a sense for the variability of wood-burning fuels across the region and how well one volatility distribution would be at simulating their emissions? Are there varying practices for controlling emissions from chimneys or flues that would have an impact on the particle fraction from these sources?

5. The authors make the point that the winter time conditions are not favorable for oxidative aging of SOA or high formation of SOA from VOCs. However, the measurement data show relatively high O:C, out of reach of the model sensitivity cases. Can the authors demonstrate the model's performance for relevant gas-phase oxidants to eliminate that as a factor?

6. I would urge the authors to consider adding more analysis of the relationship between model error and individual sources or chemical descriptions in the model. Are there correlations with other model species that would give some clues as to where the parameterizations are weak or better emissions data are needed (e.g. CO, POA, NOx, etc)? What recommendations do the authors have for future work by experimentalists and other chemical transport model efforts? What pieces of the model description need the most work? One conclusion that comes out is that the results are more sensitive to the volatility distribution than the aging mechanism. I wonder if the authors could emphasize this point as an area in need of further research? Does more work need to be done on constraining the volatility, or on representing the diversity of wood burning fuels and conditions that exist?

**Minor Issues/Typos/Suggestions**

1. Page 1, line 10: Suggest replacing "whatever the parameterizations" with "in all parameterizations tested".
2. Page 2, line 3: Suggest replacing "primary fraction originates" with "primary fraction originates mostly"
3. Page 2, line 5: evidences should be evidence
4. Page 2, line 8: I think the generally acknowledged IVOC range includes $10^3$-$10^6$ while SVOCs are $0.1$-$10^3$.
5. Page 1, line 16: Add "precursors" to read "main anthropogenic VOC precursors".
6. Page 3, lines 19-22: The 2D-VBS can also accommodate oligomerization pathways, although most transport models don't take it into account.
7. Page 3, line 23: suggest rewording to "scheme that accounts for multigenerational ageing, including functionalization and fragmentation, and that…"
8. Page 3, line 35: Recommend the authors add more description of what the non-traditional VOCs are. In the past, the word nontraditional has been used to identify SOA from IVOCs and SVOC vapors. I was confused at first, but see from the sensitivity case descriptions that these NTVOCs are different compounds.
9. Page 4, line 8: Are studies from 2001 and 2005 still recent? Obviously, this is the authors' call. Maybe everything after 2000 still 'feels' recent? It's certainly more recent than 1975.
10. Page 5, line 11: Are the authors using ISORROPIA v1? Version 2 includes among other things interactions with crustal species. If the model includes version 1, a statement should be added explaining either the unimportance of dust sources during the campaign, and/or the unimportance of crustal cations on organic aerosol concentrations as they are modeled here. The output of ISORROPIA will affect things like water uptake and pH, but most OA models now probably aren't sensitive to parameters like these, at least first- or second-order. Is that true for this model as well?

11. Page 5, line 15-16: The authors reference Chrit et al. (2017) for their grid configuration details, but I think it would still be useful to put it here. What is the grid resolution and layer resolution of the nested and large domains?
12. Page 5, lines 28-30: Is the total [I/S-VOC + POA] equal to 2.5 or 1.5 times the original POA? Could the authors adjust the wording of this sentence to make this clearer?
13. Page 6, line 9: remove "the" to read "at the model cell closest to the station"
14. Figure 1: Could the authors adjust the color scales so it's a bit easier to assess them in relation to each other? For example, 0.01 for the left and 0.05 for the right?
15. Page 7, line 9: Suggest changing "different parameterizations are compared" to "different parameterizations, described in the following sections, are compared".
16. Page 7, line 16: How are the saturation concentrations for the S/I-VOCs chosen? Are they from a previous study? Are they fit to something?
17. Table B1: Why is it that for the SOA vs. POA species, the enthalpies of vaporization are the same even though the molecular weights are higher, the saturation concentrations are somewhat lower and the O/C ratios are somewhat higher? I would guess the SOA species should have larger enthalpies of vaporization.
18. Tables D1 and D2 look to be repeated?
19. Page 11, line 6: Should "SOA" be "POA"?
20. Table 5: What is the uncertainty reflective of? One standard deviation?
21. Page 20, line 24: The authors have cited May et al. 2013a (biomass burning emissions) twice.
22.

---

## Author Comment (AC1) · 6 Nov 2018

In this work, Chrit et al. used the air quality model Polyphemus to describe the organic aerosol formation and properties (notably oxidation state) at a measurement site in Corsica during the winter campaign of 2014. The OA concentrations are well simulated by the model, however, their oxidation state is systematically underestimated compared to observations. They also stress the importance of an accurate characterization of emissions since they found that the volatility distribution at emissions is the prime factor that control the simulated OA concentration levels. Overall, the manuscript is well written and scientifically sound. I recommend this study for publication after taking the following comments into account.

**Specific comments:**

1. Page 1, lines 15-16: What is the difference between oxidation and oxygenation state of OA? If there is no difference please remove the term "oxygenation state" from this sentence

There is a difference between the oxidation and oxygenation represented by OM:OC and O:C ratios respectively. According to Gilardoni et al. 2009 and Kroll et al. 2011, the organic mass to organic carbon ratio (OM:OC) is an index of the contribution of hetero-atoms (O, H, S, N, …) to the organic mass: chemically processed and aged particles are expected to have higher OM/OC ratio compared to freshly emitted and unprocessed aerosols. However, the oxygen to carbon ratio (O:C) indicates only the contribution of oxygen to organic molecules and the ability of carbon atoms to form bonds with oxygen. This difference is now detailed in the introduction (see reply to comment 6).

2. Page 1, lines 16-17: Why do you assume that only the multigenerational ageing of the residential heating OA can improve substantially the results? Is this the dominant sector in the area? What about the multigenerational ageing of OA from other sources?

We thought that multigenerational ageing applied to I/S-VOCs from the residential heating sector may be more efficient at producing oxidized SOA, because the primary aerosols are more oxidized than those of other sectors. However, as stressed by the reviewer, it is more accurate to stress that multigenerational ageing of all sectors does not improve the results, as shown by the simulations performed. Therefore, The sentence "The observed organic oxidation and oxygenation states are strongly under-estimated in all simulations, even when a recently developed parameterization for modeling the ageing of I/S-VOC from residential heating is used." is replaced by "The observed organic oxidation and oxygenation states are strongly under-estimated in all simulations, even when multigenerational ageing of I/S-VOCs from all sectors is modeled."

3. Page 2, line 7: Please replace OA with POA.
"OA" is replaced by "POA" in the revised paper.

4. Page 2, line 3: You can also add the work of Jathar et al. (2014) and Tsimpidi et al. (2017).
These works are added to this sentence in the revised paper.

5. Page 2, line 30: You can also add the work of van der Gon et al. (2015)
This work is added to the revised paper.

6. Page 4, line 4: What is the difference between highly oxidized and highly oxygenated OA?

As explained in the reply to comment 1, oxidized and oxygenated are different. The following sentences are added at the beginning of the paragraph l7, p4: "The oxidation state is represented by the organic mass to organic carbon ratio (OM:OC). According to Gilardoni et al. 2009 and Kroll et al. 2011, OM:OC is an index of the contribution of hetero-atoms (O, H, S, N, …) to the organic mass: chemically processed and aged particles are expected to have higher OM/OC ratio compared to freshly emitted and unprocessed aerosols. The oxygenation state is represented by the oxygen to carbon ratio (O:C). It indicates the contribution of oxygen to organic molecules and the ability of carbon atoms to form bonds with oxygen."

7. Page 4, line 8: These studies are not recent. Please add more recent studies, e.g. (Aiken et al., 2008; Tost and Pringle, 2012; Canagaratna et al., 2015; Tsimpidi et al., 2018)
These more recent studies are added to the revised paper.

8. Page 5, section 2.1: OA formation from alkanes, olefins, S/I-VOC from open biomass burning, and marine OA is missing from the model setup. Can you please add a comment on their potential importance in the examined area?

OA from alkanes and olefins are not accounted for in the model setup because their emissions are very low and highly uncertain (Roest and Shade, 2017, Hajbabaei et al. 2012). Long-chain alkanes may also be included in I/S-VOCs. I/S-VOCs from open biomass burning and fires are not accounted for in the model setup because we are dealing with the OA formation during winter, and their contribution may be low. Marine OA contribute up to only 2% of OA according to Chrit et al. 2017 in summer. Its contribution during winter time may therefore be negligible.

9. Page 5, line 13: Please change „inorganics and inorgancs" with „inorganics and organics".
"inorganics and inorganics" are replaced by "inorganics and organics" in the revised paper.

10. Page 5, line 14: Please add a reference for the algorithm.
The reference for this algorithm is added in the revised paper "… moving diameter algorithm (Jacobson, 1997)".

11. Page 5, line 15: According to the presented results (e.g., Fig. 3), the simulation lasts until 2nd of April.
"01 April" in this sentence is replaced by "02 April" in the revised paper.

12. Page 5, line 16: It would be convenient if you can state the spatial resolution used here.

The spatial resolution and the vertical resolution used here are added to the revised paper: "…in Chrit et al. (2017). The spatial resolutions used for the European and Mediterranean domains are 0.5°x0.5° and 0.125°x0.125° along longitude and latitude. 14 vertical levels are used in this study for both domains from the ground to 12 km. The heights of the cell interfaces are 0, 30, 60, 100, 150, 200, 300, 500, 750, 1000, 1500, 2400, 3500, 6000 and 12 000 m. Boundary conditions…".

13. Page 5, lines 23-24: Please remove the sentence: "Other sea-salt…. not modelled"
This sentence is removed from the revised paper.

14. Page 5 line 34: How much is this constant factor (RRH)? And how much is the constant factor R stated later in the text?

Both R and RRH are equal to 1.5 in the reference simulation, and they vary in the sensitivity simulations, as summarized in Table 3. For clarity, P6, l6, the following sentence "Different approaches will also be used to represent the ageing of I/S-VOC, as described in section 3." Is replaced by "Different estimations of R and R_RH will be used, as well as different approaches to represent the ageing of I/S-VOC (section 3). "

15. Page 6, section 2.2: PMF analysis results would be very useful for comparison with your model results. Are they available at the Ersa site? If so, please add this comparison.
Unfortunately, PMF analysis results are not available at Ersa.

16. Page 6, line 10: Are these coordinated the center of the model cell? Does the dimension of the model cell include the coordinates of the station mentioned above?

Yes, these are the coordinates of the center of the model cell. The dimension of the model cell does not include the coordinates of the station. Therefore, we compare the measured data at Ersa with the simulated ones at the center of the cell the closest to the station.

17. Page 6, line 10: Please define the abbreviation ACSM
ACSM abbreviation is defined in the introduction (page 2 line 23).

18. Page 6, line 12: Did you compare the measurements with the model results with size sections from 0.056 to 1.0 (as they appear in the previous page)? Please clarify.
The measurements are compared to model results between 0.01 and 1$\mu$m. This sentence is added to section 2.2.

19. Page 6, line 14: Please correct "he" with "the"
"he" is replaced by "the" in the revised paper.

20. Page 7, section 3: What about traditional VOCs? Are they subject to photochemical ageing?

The traditional VOCs considered are toluene and xylene. These VOCs are subject to photochemical ageing based on chamber measurements. They are modelled with a one-step oxidation scheme. Their contribution to OA is low as specified in the introduction p3. In the introduction p3, the sentence
"In winter, when anthropogenic emissions impact the most air quality, anthropogenic emissions such as toluene and xylenes may also form SOA, although they may be less efficient than I/S-VOC (Couvidat et al., 2013a)" is replaced by "In winter, when anthropogenic emissions impact the most air quality, anthropogenic emissions such as toluene and xylenes may also form SOA, although they may be much less efficient than I/S-VOC (Couvidat et al., 2013a, Sartelet et al. 2018)"

21. Page 8, section 3.3: How do you treat OA from sources other than residential heating in this case? Do they follow the oxidation scheme described in section 3.2? In that case, can you justify why you use a different oxidation scheme especially for residential heating and not for other sources?

Please see section 3.5 where different sensitivity studies are performed using different oxidation scheme for I/SVOCs from residential heating but also for I/S-VOCs from other sources. The residential heating sector was studied separately because it makes a large part of I/S-VOCs emissions, but also because its emissions are more oxidized and oxygenated than the ones from other sources like traffic.

22. Page 8, line 11: The carbon number should decrease and oxygen number increases, please correct.

The sentence "… secondary surrogates increases and decreases respectively…" is replaced by "…secondary surrogates decreases and increases respectively … " in the revised paper.

23. Page 8, Section 3.4:Can you provide the actual emission rates (e.g., in Tn yr -1 ) of your OA precursor emissions (i.e., VOCs, NTVOCs, I/S – VOCs from different types of sources)?

A table of emission rates of OA precursors averaged over the Mediterranean domain and over the simulation period is added to section 3.4 of the revised paper.

| OA precursor | Emission rate ($\mu g.m^{-2}s^{-1}$) |
| --- | --- |
| VOCs from biogenic and anthropogenic sources | 0.0314 |
| NTVOCs | 0.0062 |
| I/S-VOCs from residential heating | 0.0013 |
| I/S-VOCs from other sources | 0.0030 |

24. Page 9, Tables 1 and 2: Please improve the quality of the tables. For example, you should include names of surrogate species that you assign these numbers, names of sensitivity tests, and what these numbers express (i.e., emission factors, O:C, OM/OC should not be stated only in the caption but also inside the tables).

The name of primary surrogates are added between brackets to the table 1 according to their volatility coefficient and the definition of the numbers is also added to both tables 1 and 2 of the revised paper. Furthermore, a Table is added to section 3.5 to summarize the sensitivity tests.

| Sensitivity study of: | Simulations to be compared |
| --- | --- |
| the impact of the volatility distribution of emissions | S1, S2 |
| the impact of the ageing scheme | S3, S2 |
| the impact of NTVOCs | S4, S2 |
| the impact of the I/S-VOCs/POA ratio | S5, S2 and S6,S4 |

25. Page 10, line 11: This is not very clear. You apply a factor of 4 in the initial emission inventory, and then, on top of that you apply a factor of 4.75 to account for the NTVOC (which are not part of your S/I-VOC). Can you please clarify and justify your hypothesis of such high additional emissions?

Yes, this is what we did, following the papers published by Ciarelli et al. (2016, 2017). In fact, it is a sensitivity test to investigate how the concentrations compare to measurements using these published emission factors.

26. Page 10, line 17: Please add in the sentence the average OM concentrations over these cities. Likewise, provide average concentrations for other mentioned areas (e.g., Ersa) later throughout the text.

The average OM concentration simulated using S4 over the mentioned cities is added to the revised paper.

27. Page 10, line 19: Why do you focus only in these two simulations?
We focus on these two simulations because they are the ones that simulate better the $OM_1$ concentrations.

28. Page 10, line 19: Do you mean in both simulations (instead of "in all simulations")?
No, we mean all simulations, even though we do not show their composition.

29. Page 11, Figure 2: Please increase the font size of the fractions. Also, the fractions over the dark blue are not clear.
This figure is more readable in the revised paper.

30. Page 12, line 1: OA is already defined.
Organic aerosol is removed from this sentence in the revised paper.

31. Is this an assumption or did you actually check that you have a false rain episode in your model?
It is an assumption because we do not have rain observations at Ersa.

32. Page 15, line 5: OA is already defined.
Organic aerosol is removed from this sentence in the revised paper.

33. Page 15, Section 6: Can you comment on the importance of marine OA in your domain? Is this type of OA identified by measurements at Ersa site?

Chrit et al. 2017 examined the influence of marine OA during summer and found that the contribution of marine OA to OA concentrations over Ersa is lower than 2%. The contribution during the winter would be even lower.

34. Appendix A, table A1: The definitions are not clearly readable.

This table is more clearly readable in the revised paper.

References:

Aiken, A. C., Decarlo, P. F., Kroll, J. H., Worsnop, D. R., Huffman, J. A., Docherty, K. S., Ulbrich, I. M., Mohr, C., Kimmel, J. R., Sueper, D., Sun, Y., Zhang, Q., Trimborn, A., Northway, M., Ziemann, P. J., Canagaratna, M. R., Onasch, T. B., Alfarra, M. R., Prevot, A. S. H., Dommen, J., Duplissy, J., Metzger, A., Baltensperger, U., and Jimenez, J. L.: O/C and OM/OC ratios of primary, secondary, and ambient organic aerosols with high-resolution time-of-flight aerosol mass spectrometry, Environmen. Sci. & Technol., 42, 4478-4485, 2008.

Canagaratna, M. R., Jimenez, J. L., Kroll, J. H., Chen, Q., Kessler, S. H., Massoli, P., Ruiz, L. H., Fortner, E., Williams, L. R., Wilson, K. R., Surratt, J. D., Donahue, N. M., Jayne, J. T., and Worsnop, D. R.: Elemental ratio measurements of organic compounds using aerosol mass spectrometry: characterization, improved calibration, and implications, Atmospheric Chemistry and Physics, 15, 253-272, 2015.

Gilardoni, S., Liu, S., Takahama, S., Russell, L.M., Allan, J.D., Steinbrecher, R., et al. Characterization of Organic Ambient Aerosol during MIRAGE 2006 on Three Platforms. Atmospheric Chemistry and Physics, 9, 5417-5432, https://doi.org/10.5194/acp-9-5417-2009, 2009.

Hajbabaei M, Johnson KC, Okamoto R, et al. Evaluation of the impacts of biodiesel and second generation biofuels on NO(x) emissions for CARB diesel fuels. Environ. Sci . Technol. 2012;46:9163–9173.

Jathar, S. H., Gordon, T. D., Hennigan, C. J., Pye, H. O. T., Pouliot, G., Adams, P. J., Donahue, N. M., and Robinson, A. L.: Unspeciated organic emissions from combustion sources and their influence on the secondary organic aerosol budget in the United States, Proceedings of the National Academy of Sciences of the United States of America, 111, 10473-10478, 2014.

Kroll, J. H., Donahue, N. M., Jimenez, J. L., Kessler, S. H., Canagaratna, M., Wilson, K. R., Altieri, K. E., Mazzoleni, L. R., Wozniak, A. S., Bluhm, H., Mysak, E. R., Smith, J. D., E., K. C., and Worsnop, D. R.: Carbon oxidation state as a metric for describing the chemistry of atmospheric organic aerosol, Nature Chem., 3, 133–139, https://doi.org/10.1038/NCHEM.948, 2011.

Roest, G., & Schade, G. (2017). Quantifying alkane emissions in the Eagle Ford shale using boundary layer enhancement. Atmospheric Chemistry and Physics, 17(18), 11,163–11,176. https://doi.org/10.5194/acp-17-11163-2017

Sartelet K., Zhu S., Moukhtar S., André M., André J.M., Gros V., Favez O., Brasseur A., Redaelli M. (2018), Emission of intermediate, semi and low volatile organic compounds from traffic and their impact on secondary organic aerosol concentrations over Greater Paris. Atmos. Env., 180, 126-137, doi:10.1016/j.atmosenv.2018.02.031.

Tost, H. and Pringle, K. J.: Improvements of organic aerosol representations and their effects in large-scale atmospheric models, Atmospheric Chemistry and Physics, 12, 8687- 8709, 2012.

Tsimpidi, A. P., Karydis, V. A., Pandis, S. N., and Lelieveld, J.: Global-scale combustion sources of organic aerosols: sensitivity to formation and removal mechanisms, Atmospheric Chemistry and Physics, 17, 7345-7364, 2017.

Tsimpidi, A. P., Karydis, V. A., Pozzer, A., Pandis, S. N., and Lelieveld, J.: ORACLE 2-D (v2.0): An efficient module to compute the volatility and oxygen content of organic aerosol with a global chemistry - climate model, Geosci. Model Dev. Discuss., 2018, 1-45, 2018.

van der Gon, H., Bergstrom, R., Fountoukis, C., Johansson, C., Pandis, S. N., Simpson, D., and Visschedijk, A. J. H.: Particulate emissions from residential wood combustion in Europe revised estimates and an evaluation, Atmospheric Chemistry and Physics, 15, 6503-6519, 2015.

---

## Author Comment (AC2) · 6 Nov 2018

Comment on "Modeling organic aerosol concentrations and properties during winter 2014 in the northwestern Mediterranean region" by Chrit et al.

acp-2018-149

**General Comments:**

Chrit et al. have deployed the Polyphemus platform with expanded techniques for simulating organic aerosol formation and aging, particularly from residential heating sources, and applied it to the CharMEx campaign in winter 2014. The study design is generally sound and the sensitivity choices are informative. There is also adequate reporting of direct results. However, I found there to be a lack of further diagnosis or interpretation of results considering the amount of data the authors would have access to from the model. Considering this, I think the paper would be better suited in its current form for GMD, although I think it is of acceptable quality and scope for publication in ACP, given that the authors address specific concerns below.

**Specific Comments**

1.  Section 3.4: I did not understand how profiles 1 and 2 are allocated to individual emissions. It should be made clearer why profiles 1 and 2 can be mismatched for volatility and O:C? Usually, I think of these as tied together by the molecular weight (i.e. carbon number) assumed for each model species of a given volatility. Are the assignments made using some feature from the emissions inputs, or are individual sectors assigned profile 1 or 2 based on some knowledge of their emissions (e.g. waste burning goes with profile 2, offroad diesel goes with profile number 1, etc)? If the latter, can the authors include a table that identifies these assignments? If the former, can the authors better describe what parameters and algorithm are used to make the assignments?

Profiles 1 and 2 are used to allocate emissions of I/S-VOCs into model surrogate species. This allocation depends on the emission sectors. In the profiles, volatility and O:C can mismatch for two reasons: the volatility range spans by one model species is quite large and the chemical compounds may have different functional groups. The profiles 1 and 2 are based on chamber measurements performed for different emission sectors: profile 2 for I/S-VOCs from residential heating sector and profil 1 for I/S-VOCs from traffic.  For clarity, the beginning of section 3.4 was rewritten as follows:
"Emissions of I/S-VOCs are allocated into the surrogate compounds detailed in the above sections using emission distribution profiles, which are based on chamber measurements. The distribution of the emission profiles as a function of volatility (saturation concentration) is detailed in Table 1. Two emission profiles are used. The first one corresponds to the measurements of May et al. (2013a) for biomass burning, and it is similar to the emission profile used by Couvidat et al. (2012) for all sectors and by Ciarelli et al. (2017b) for residential heating. The second emission profile corresponds to an average of emission distributions from gasoline and diesel vehicles measured by May et al. (2013b, c), and it is used in Koo et al. (2014). Here, the volatility emission distributions are assigned to a profile number (equal to 1 or 2), depending on whether the volatility profile is similar to the profile from biomass burning emissions of May et al. (2013b) (profile number 2) or whether it is similar to the profile from vehicle emissions of May et al. (2013c) and May et al. (2013a) (profile number 1). As shown in Table 1, the emitted I/S-VOC are less volatile in the profile 1 than in the biomass-burning volatility distribution (profile 2).
Depending on the emission sector, the OM:OC and O:C ratios of the emitted surrogates may differ. For most sectors, such as traffic, the OM:OC and O:C ratios are assumed to be low (OM:OC is equal to 1.3 in Couvidat et al. 2012). However, for residential heating, the emissions may be more oxidized. The scheme of Ciarelli et al. (2017b) assumes higher OM:OC and O:C rations, as described in Table 2. Here, the OM:OC and O:C ratios are assigned to a profile number (equal to 1 or 2), depending on whether the ratios are similar to the profile from biomass burning emissions of Ciarelli et al. (2017b) (profile number 2) or whether they are lower (profile number 1)."
In the simulation, the assignment to profile 1 or 2 is identified in Table 3.

2.  After going back to Ciarelli et al. (2017b), I am not convinced they included additional IVOCs, consistent with those being added in this simulation, in their parameterizations. It seems from Table 1 in that paper, that the authors included NTVOCs and also evaporated the existing POA into SVOC and IVOC bins. But I do not think they considered an additional IVOC category. Given this, I am not surprised that simulations here which include additional IVOCs and NTVOCs (S4 and S6) tend to over predict measurements. I would suggest the authors perform at least one run with R_RH set to 0 for residential heating sources and NTVOC turned on. This will probably look a lot like S5 so if the authors want to adjust the explanation of their cases to avoid doing more simulations, I think that is okay, but some detailed explanation should be added (i.e. R_RH could be defined as

adding NTVOC). Note this approach would not be perfect, because the SOA yields for the IVOCs would differ from those Ciarelli et al. (2017b) derived for NTVOC.

In Ciarelli et al. (2017b), NTVOCs have a saturation concentration of $10^6$ µg m$^{-3}$ falling with the IVOC saturation concentration range limit. These NTVOCs probably include some VOCs and some IVOCs. However, modelling IVOCs and SVOC emissions by multiplying POA by a factor accounts for the fact that in the emission inventory the gas-phase of I/S-VOCs is not given. We agree with the reviewer that some IVOCs are probably counted twice if NTVOCs are added to the emissions, as well as the factor to estimate I/S-VOCs from POA. Several changes are therefore made to the revised paper:

The sentence "but they are slightly over-estimated when the ageing of NTVOC is taken into account" is removed from the abstract.

On page 3, the sentence "Ciarelli et al. (2017b) modified the approach of Koo et al. (2014) by adding non traditional VOC (NTVOC) that have a limit saturation concentration between VOC and IVOC." is replaced by "Ciarelli et al. (2017b) modified the approach of Koo et al. (2014) by considering non traditional VOC (NTVOC) that have a limit saturation concentration in the low range of IVOCs."
On page 10, at the end of line 7, the following sentence is added: "Even though NTVOCs are added, emissions of I/S-VOCs as modeled by the factor R_RH are kept."

On page 11, at the end of line 16, the following sentences are added: "Because I/S-VOC emissions as modeled by the factor R_RH are kept in those simulations, the IVOCs forming SOA may have been counted twice by adding NTVOCs, explaining the over-estimation."
The sentences lines 18-20 on page 13 are removed.

On page 20, the words " particularly those with NTVOC" are removed.
On page 20, the sentence "All the simulations tend to under-estimate the OA concentrations at Ersa, except for the two simulations where NTVOC are taken into account, which, however, over-estimate the OA concentrations." is replaced by "All the simulations tend to under-estimate the OA concentrations at Ersa, except for the two simulations where NTVOC emissions are added to I/S-VOC emissions. These simulations over-estimate the OA concentrations, because some IVOC emissions are counted twice."

3. Are the IVOCs from residential heating assumed to be the same composition (i.e. same SOA yields) as those from vehicle sources? If so, what is this based on?

In the one-step oxidation scheme, IVOCs from residential heating are assumed to have the same SOA yield as those from vehicle sources. This assumption is commonly made in 3D models (e.g. Couvidat et al. 2012). It is based on the work of Shrivastava et al. (2006), who show a very similar dilution curve behavior between diesel exhaust and wood smoke.

4. Do the authors have a sense for the variability of wood-burning fuels across the region and how well one volatility distribution would be at simulating their emissions? Are there varying practices for controlling emissions from chimneys or flues that would have an impact on the particle fraction from these sources?

The volatility distribution of May et al. (2013) used for the wood burning emissions that is based on fitting data from thermodenuder measurements of the burning of 19 wood types. They found that the overall partitioning behavior of all the biomass fuel emissions considered in their study is similar enough to be represented in the model by one parameterization. Furthermore, we do not have data on the wood-burning fuels used across the region. Knowledge about wood-burning fuel may be complicated by the fact that the wood-burning fuel used may differ from official recommendations. The variability of wood-burning fuels may however be more important for very low-volatility emissions, which are difficult to measure. This point was added to the conclusion (see reply to comment 6).

5. The authors make the point that the winter time conditions are not favorable for oxidative aging of SOA or high formation of SOA from VOCs. However, the measurement data show relatively high O:C, out of reach of the model sensitivity cases. Can the authors demonstrate the model's performance for relevant gas-phase oxidants to eliminate that as a factor?

Only ozone was measured at ERSA. We do not have other oxidants' measurements.

The comparison of modeled and measured concentrations of ozone between January 21and February 24 is added to the revised paper. This figure shows that the model tends to underestimate ozone concentrations (the modeled and measured mean concentrations are 46.2 and 68.0 µg m$^{-3}$). This suggests that the underestimation of the O:C ratio may be due to an underestimation of oxidants' concentrations and secondary aerosol formation. However, the O:C ratio is underestimated even during the days where ozone is well modeled.
These sentences are added to the revised paper in section 5, and the following sentences are added to the conclusion. : «… OM:OC and O:C ratios are underestimated at Ersa in all simulations. As ozone tends to be underestimated in the model compared to the measurements, the underestimation of the OM:OC and O:C ratios might partly be due to an underestimation of oxidants concentrations and secondary aerosol formation. »

6. I would urge the authors to consider adding more analysis of the relationship between model error and individual sources or chemical descriptions in the model. Are there correlations with other model species that would give some clues as to where the parameterizations are weak or better emissions data are needed (e.g. CO, POA, NOx, etc)? What recommendations do the authors have for future work by experimentalists and other chemical transport model efforts? What pieces of the model description need the most work? One conclusion that comes out is that the results are more sensitive to the volatility distribution than the aging mechanism. I wonder if the authors could emphasize this point as an area in need of further research? Does more work need to be done on constraining the volatility, or on representing the diversity of wood burning fuels and conditions that exist?

Based on the paper of May et al. (2013), representing the diversity of wood-burning fuels does not seem to be influence the partitioning between gas and particle, although the emissions of low volatility compounds, which are not well characterized, may differ.
A paragraph emphasizing the future work and areas where the model needs more improvement is added at the beginning of the paragraph at line 23 in the conclusion of the discussion paper. "Because the volatility distribution at the emission is the parameter influencing the most the concentrations, further experimental research should therefore focus on characterizing it for the different sectors. The emissions and formation of very low-volatility compounds should also be further investigated to represent the aerosol characteristics observed."

**Minor Issues/Typos/Suggestions**

1. Page 1, line 10: Suggest replacing "whatever the parameterizations" with "in all parameterizations tested".
"Whatever the parameterizations" is replaced by "in all parameterizations" in the revised paper.

2. Page 2, line 3: Suggest replacing "primary fraction originates" with "primary fraction originates mostly"
"primary fraction originates" is replaced by "primary fraction originates mostly" in the revised paper.

3. Page 2, line 5: evidences should be evidence
"evidences" is replaced by "evidence" in the revised paper.

4. Page 2, line 8: I think the generally acknowledged IVOC range includes $10^3$ -$10^6$ while SVOCs are 0.1-$10^3$.

Theses ranges are corrected in the revised paper. "… (IVOC) (with saturation concentration C∗ in the range $10^4$ -$10^6$ µg m$^{-3}$), semi-volatile organic compounds (SVOC) (with saturation concentration C ∗ in the range 0.1-$10^4$ µg m$^{-3}$), or low-volatility …" is replaced by "… (IVOC) (with saturation concentration C∗ in the range $10^3$ -$10^6$ µg m$^{-3}$), semi-volatile organic compounds (SVOC) (with saturation concentration C ∗ in the range 0.1-$10^3$ µg m$^{-3}$), or low-volatility …" in the revised paper.

5. Page 1, line 16: Add "precursors" to read "main anthropogenic VOC precursors".

This expression is actually in page 2 line 16.
"… main anthropogenic VOC …" is replaced by "… main anthropogenic VOC precursors …" in the revised paper.

6. Page 3, lines 19-22: The 2D-VBS can also accommodate oligomerization pathways, although most transport models don't take it into account.

The sentence " …taking into account two competing processes: functionalization and fragmentation (Donahue et al., 2012). …" is replaced in the revised paper by "taking into account three competing processes: functionalization, oligomerization and fragmentation (Donahue et al., 2012).".

7. Page 3, line 23: suggest rewording to "scheme that accounts for multigenerational ageing, including functionalization and fragmentation, and that…"

The sentence "… scheme that accounts for fragmentation, functionalization and multigenerational ageing, and that represents …" is replaced in the revised paper by "scheme that accounts for multigenerational ageing, including functionalization, oligomerization and fragmentation, and that represents …"

8. Page 3, line 35: Recommend the authors add more description of what the non-traditional VOCs are. In the past, the word nontraditional has been used to identify SOA from IVOCs and SVOC vapors. I was confused at first, but see from the sensitivity case descriptions that these NTVOCs are different compounds.

The following sentence is added to clarify the definition of NTVOCs in the revised paper. "… adding non traditional VOCs (NTVOCs).  They are VOCs or IVOCs, not usually taken into account in CTMs, and with a saturation concentration in the low-range of IVOCs . Ciarelli et al. (2016) identified these NTVOCs as phenol, m-, o-, p-cresol, m-, 15 o-, p-benzenediol/2-methylfuraldehyde, dimethylphenols, guaiacol/methylbenzenediols, naphthalene, 2-methylnaphthalene/1- methylnaphthalene, acenaphthylene, syringol, biphenyl/acenaphthene and dimethylnaphthalene".

9. Page 4, line 8: Are studies from 2001 and 2005 still recent? Obviously, this is the authors' call. Maybe everything after 2000 still 'feels' recent? It's certainly more recent than 1975.

The sentence "… , recent studies (Turpin and Lim, 2001; El-Zanan et al., 2005) show …" is replaced by "…, numerous studies (Turpin and Lim, 2001; El-Zanan et al., 2005; Aiken et al., 2008, Couvidat et al., 2012, Tost and Pringle, 2012, Canagaratna et al., 2015, Tsimpidi et al., 2018) show …".

10. Page 5, line 11: Are the authors using ISORROPIA v1? Version 2 includes among other things interactions with crustal species. If the model includes version 1, a statement should be added explaining either the unimportance of dust sources during the campaign, and/or the unimportance of crustal cations on organic aerosol concentrations as they are modeled here. The output of ISORROPIA will affect things like water uptake and pH, but most OA models now probably aren't sensitive to parameters like these, at least first- or second-order. Is that true for this model as well?

The ISORROPIA version used in this study is ISORROPIA v1 (Nenes et al. 1998). Crustal cations are not taken into account in this work, although they may affect water uptake and pH. However, as a first approximation, I/S-

VOCs are assumed to be hydrophobic, and therefore their concentrations would not be influenced by crustal species.

11. Page 5, line 15-16: The authors reference Chrit et al. (2017) for their grid configuration details, but I think it would still be useful to put it here. What is the grid resolution and layer resolution of the nested and large domains?

The spatial resolution and the vertical resolution used here are added to the revised paper: "…in Chrit et al. (2017). The spatial resolutions used for the European and Mediterranean domains are 0.5ºx0.5º and 0.125ºx0.125º along longitude and latitude. 14 vertical levels are used in this study for both domains from the ground to 12 km. The heights of the cell interfaces are 0, 30, 60, 100, 150, 200, 300, 500, 750, 1000, 1500, 2400, 3500, 6000 and 12 000 m. Boundary conditions…".

12. Page 5, lines 28-30: Is the total [I/S-VOC + POA] equal to 2.5 or 1.5 times the original POA? Could the authors adjust the wording of this sentence to make this clearer?

The total [I/S-VOC + POA] is equal to 2.5 times the original POA. For clarity, the sentence "I/S-VOC gas-phase emissions are estimated from the POA emissions from residential heating by multiplying them by a constant factor assumed to be 1.5 in the default simulation." is replaced by "I/S-VOC gas-phase emissions are estimated from the POA emissions from residential heating by multiplying them by a constant factor assumed to be 1.5 in the default simulation. The total (gas + particle) I/S-VOCs     is therefore equal to 2.5 the original POA."

13. Page 6, line 9: remove "the" to read "at the model cell closest to the station"
"the" is removed from that sentence.

14. Figure 1: Could the authors adjust the color scales so it's a bit easier to assess them in relation to each other? For example, 0.01 for the left and 0.05 for the right?
The color scale of this figure is adjusted in the revised paper.

15. Page 7, line 9: Suggest changing "different parameterizations are compared" to "different parameterizations, described in the following sections, are compared".

"different parameterizations are compared" is replaced by "different parameterizations, described in the following sections, are compared" in the revised paper.

16. Page 7, line 16: How are the saturation concentrations for the S/I-VOCs chosen? Are they from a previous study? Are they fit to something?

These saturation concentrations for the I/S-VOCs are chosen to fit the curve of dilution of POA from diesel exhaust of Robinson et al. (2007) with three molecules. This point is added in the revised paper: "… different volatilities chosen to fit the dilution curve of POA from diesel exhaust of Robinson et al. (2007) and characterized by their saturation concentrations (0.91, 86.21 and 3225.80 µg m$^{-3}$ respectively) …".

17. Table B1: Why is it that for the SOA vs. POA species, the enthalpies of vaporization are the same even though the molecular weights are higher, the saturation concentrations are somewhat lower and the O/C ratios are somewhat higher? I would guess the SOA species should have larger enthalpies of vaporization.

The enthalpies of vaporizations are assumed to be the same for SOA as for POA because of lack of experimental data. It is difficult to estimate what the enthalpy of vaporization of SOA should be. A recent study of Majdi et al. acpd, (2018) found that the sensitivity of AOS concentrations formed from fire emissions to variations in the modeled enthalpy of vaporization is low compared to other sensitivities, such as the ageing scheme.

18. Tables D1 and D2 look to be repeated?
Yes, the table D2 is removed from the revised paper.

19. Page 11, line 6: Should "SOA" be "POA"?
Yes, "… 31% of SOA from …" is replaced by "… 31% of POA from …"  in the revised paper.

20. Table 5: What is the uncertainty reflective of? One standard deviation?

Yes, it is the standard deviation to the measurements.

21. Page 20, line 24: The authors have cited May et al. 2013a (biomass burning emissions) twice. The second A in the name of May is removed from that reference in the revised paper.

---

## Author Response (AR2)

**Co-Editor Decision: Publish subject to minor revisions (review by editor)** (15 Nov 2018) by Matthias Beekmann
Comments to the Author:
Globally, the authors answered in a satisfying way to the referees remarks. I recommend publication in ACP, which has the advantage over GMD to have a special section dedicated to the ChArMEX project.
There are several minor remarks that still need attention before publication.

Referee 1 , page 10, line 11:
"This is not very clear. You apply a factor of 4 in the initial emission inventory, and then, on top of that you apply a factor of 4.75 to account for the NTVOC (which are not part of your S/I-VOC). Can you please clarify and justify your hypothesis of such high additional emissions?"
Authors response:
"Yes, this is what we did, following the papers published by Ciarelli et al. (2016, 2017). In fact, it is a sensitivity test to investigate how the concentrations compare to measurements using these published emission factors."
Editors recommendation:
Could the authors please explain in a few words how Ciarelli et al. obtained this factor of 4.75 ?
Ciarelli et al. (2017b) measured non traditional VOC (NTVOC) that have a limit saturation concentration in the low range of IVOCs. In the reply to the point 8 of the reviewer 2, we added in the introduction p3 l30 : "… adding non traditional VOCs (NTVOCs). They are VOCs or IVOCs, not usually taken into account in CTMs, and with a saturation concentration in the low-range of IVOCs . Ciarelli et al. (2016) identified these NTVOCs as phenol, m-, o-, p-cresol, m-, 15 o-, p-benzenediol/2-methylfuraldehyde, dimethylphenols, guaiacol/methylbenzenediols, naphthalene, 2-methylnaphthalene/1- methylnaphthalene, acenaphthylene, syringol, biphenyl/acenaphthene and dimethylnaphthalene". For clarity on how the factor 4.75 is obtained, the sentence above was modified to

"… adding non traditional VOCs (NTVOCs). They are VOCs or IVOCs, not usually taken into account in CTMs, and with a saturation concentration in the low-range of IVOCs . Following Bruns et al. (2016), Ciarelli et al. (2017b) estimated these NTVOCs, using chamber experiments, as the mixture of phenol, m-, o-, p-cresol, m-, 15 o-, p-benzenediol/2-methylfuraldehyde, dimethylphenols, guaiacol/methylbenzenediols, naphthalene, 2-methylnaphthalene/1- methylnaphthalene, acenaphthylene, syringol, biphenyl/acenaphthene and dimethylnaphthalene. Furthermore, they estimated the ratio NTVOC/SVOC, where SVOC is the primary semi-volatile organic matter, to be about 4.75."

Referee 2:

5. The authors make the point that the winter time conditions are not favorable for oxidative aging of SOA or high formation of SOA from VOCs. However, the measurement data show relatively high O:C, out of reach of the model sensitivity cases. Can the authors demonstrate the model's performance for relevant gas-phase oxidants to eliminate that as a factor?
"«… OM:OC and O:C ratios are underestimated at Ersa in all simulations. As ozone tends to be underestimated in the model compared to the measurements, the underestimation of the OM:OC and O:C ratios might partly be due to an underestimation of oxidants concentrations and secondary aerosol formation. »

I only partly agree with the referee's suggestion and the authors response. Indeed, oxidant levels as OH, O3, NO3 and Cl first trigger initial attack to VOC's or I-SVOCs. They also can trigger aging, when it is included in the model.
But: (1) O3 is only one oxidant species among others; OH during wintertime probably has other major sources than O3 photolysis. It could be over – or underestimated in the model, but uncertainty is high.

Yes, indeed. P15, after "However, the O:C ratio stays underestimated even during the days where ozone is well modeled.", the following sentence is added: "It is difficult to conclude on the underestimation of oxidants,

because measurements were not performed for other oxidants than ozone, such as OH, which probably has other sources than ozone photolysis in winter."

The sentences "As ozone tends to be underestimated in the model compared to the measurements, the underestimation of the OM:OC and O:C ratios might partly be due to an underestimation of oxidants' concentrations and secondary aerosol formation" are removed from the conclusion.

(2) the chemical scheme should much impact the oxidation state: for instance, for autooxidation reactions (included in another paper of Chrit et al.), additional oxygen is provided by molecular oxygen and this independently of oxidant levels, once the initial attack is made. So authors should please broaden the discussion.

To emphasize the impact of autoxidation, in the conclusion, the sentences :

« For the summer campaign, Chrit et al. (2017) improved the simulated OM:OC ratio by adding the formation mechanisms of both extremely-low volatile organic compounds and organic nitrate from monoterpene oxidation. Similarly, the formation of organic nitrate and highly oxygenated organic molecules (Molteni et al., 2018) from aromatic precursors should be added in order to better reproduce the observed OA oxidation/oxygenation levels. » are modifed to

« For the summer campaign, Chrit et al. (2017) improved the simulated OM:OC ratio by adding the formation mechanisms of extremely-low volatile organic compounds from the autoxidation of monoterpenes and organic nitrate from monoterpene oxidation. Similarly, the formation of organic nitrate and highly oxygenated organic molecules (Molteni et al., 2018) from the autoxidation of aromatic precursors should be added in order to better reproduce the observed OA oxidation/oxygenation levels. »

6. I would urge the authors to consider adding more analysis of the relationship between model error and individual sources or chemical descriptions in the model. Are there correlations with other model species that would give some clues as to where the parameterizations are weak or better emissions data are needed (e.g. CO, POA, NOx, etc)?
The authors did not address this specific remark of referee 2.

We identified the weak model parameterizations and uncertain emission data. As discussed in the conclusion, we believe that the discrepanties between model and observations are due to processes not accounted in the model : autoxidation and the formation of organic nitrate. The sensitivity study performed showed that we are able to model well the concentrations, but they are strongly dependent on I/SVOC emissions and volatility, which are very uncertain. That is why in the conclusion, we emphasized : « 
[revised manuscript text omitted]

Table 2 (laid out):

| Saturation Conc. | Profil 1 May et al. (2013c, d) | Profil 2 Couvidat et al. (2012) | Saturation Conc. | Profil 1 May et al. (2013c, d) | Profil 2 May et al. (2013b) |
|---|---|---|---|---|---|
| 0.9 | 0.35 [POAlP] | 0.25 [BBPOAlP] | 0.1 | 0.15 [VAP0] | 0.20 [BBPOA0] |
| | | | 1 | 0.20 [VAP1] | 0.10 [BBPOA1] |
| 86.2 | 0.51 [POAmP] | 0.32 [BBPOAmP] | 10 | 0.31 [VAP2] | 0.10 [BBPOA2] |
| | | | 100 | 0.20 [VAP3] | 0.20 [BBPOA3] |
| 3225.8 | 0.14 [POAhP] | 0.43 [BBPOAhP] | 1000 | 0.14 [VAP4] | 0.4 [BBPOA4] |

**Table 2.** Summary of the volatility distributions of the primary I/S-VOC surrogates. Saturation concentrations are expresssed in $\mu$g m$^{-3}$. For each saturation concentration and volatility coefficient, the name of the associated primary surrogate is between two square brackets.

[revised manuscript text omitted]